# RefDeblur: Blind Motion Deblurring with Self-Generated Reference Image

**Insoo Kim**                                          *insoo1.kim@samsung.com*
*AI Center, Samsung Electronics*
*Korea Advanced Institute of Science & Technology (KAIST)*

**Geonseok Seo**                                       *gunsuk.seo@samsung.com*
*AI Center, Samsung Electronics*
*Seoul National University (SNU)*

**Hyong-Euk Lee**                                      *hyongeuk.lee@samsung.com*
*AI Center, Samsung Electronics*

**Jinwoo Shin**                                        *jinwoos@kaist.ac.kr*
*Korea Advanced Institute of Science & Technology (KAIST)*

**Reviewed on OpenReview:** *https://openreview.net/forum?id=Nyewu7xztw*

## Abstract

The challenge of blind motion deblurring is often tackled via two distinct paradigms: kernel-based and kernel-free methods. Each deblurring method provides inherent strengths. Kernel-based methods facilitate generating texture-detailed sharp images by closely aligning with the blurring process. In contrast, kernel-free methods are more effective in handling complex blur patterns. Building upon these complementary benefits, we propose a hybrid framework that decomposes a non-uniform deblurring task into two simpler tasks: a uniform kernel estimation, managed by our kernel-based method, and error prediction, handled by our kernel-free method. Our kernel-based method serves to generate a reference image with realistic texture details while our kernel-free model refines the reference image by correcting residual errors with preserving texture details. To efficiently build our kernel-based model, we consider the logarithmic fourier space that facilitates estimating a blur kernel easier by simplifying the relationship between blur and sharp samples. Furthermore, the regime under using a texture-detailed reference image allows for reducing the size of our kernel-free model without compromising performance. As a result, the proposed method achieves remarkable performance on several datasets such as RealBlur, RSBlur and GoPro, and comparable performance to state-of-the-art methods with a 75% reduction in computational costs.

## 1 Introduction

Motion blur images often occur in low-light environments and scenes involving high-speed movements. They are caused by camera motion and object movement in the long exposure time. In low-light conditions, using a long exposure setting reduces noise but results in substantial blur. The goal of blind image deblurring is to restore sharp images from blur ones.

In earlier years, various deep methods have been devoted to investigating accurate blur kernel estimation (Schuler et al., 2015; Chakrabarti, 2016; Gong et al., 2017; Tran et al., 2021; Zhang et al., 2021; Carbajal et al., 2023). These kernel-based techniques prioritize estimating the blur kernels, which are then used in the deconvolution process (Fish et al., 1995; Krishnan & Fergus, 2009) to generate sharp images. Although they show promising results in certain cases, their performance is not acceptable for real-world scenarios. In fact, there are some drawbacks to kernel-based methods. First, they train with synthesized motion blur images

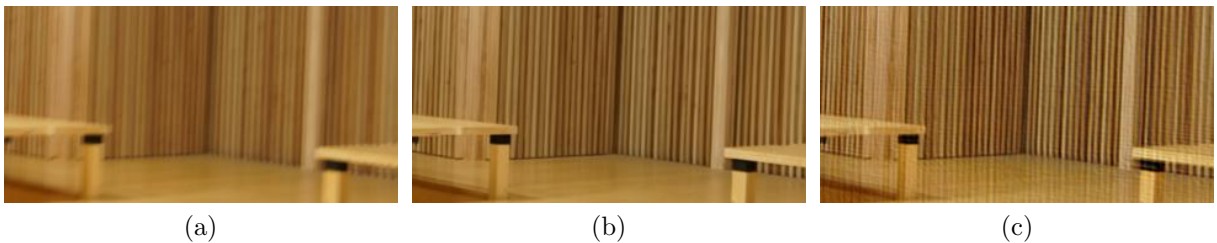

<table>
<tr><td>(a)</td><td>(b)</td><td>(c)</td></tr>
</table>

Figure 1: Visual comparison of the reference images: (a) Blur image, (b) NAFNet (Chen et al., 2022), and (c) Our kernel-based model with $\lambda = 0.01$.

that may not hold in practice. Second, estimating a blur kernel for every pixel is a huge ill-posed problem. Therefore, they may lead to unfavorable ringing artifacts due to inaccurate kernels, especially in noise and saturation pixels (Yuan et al., 2007; Whyte et al., 2014). Third, finding accurate per-pixel kernels may lead to increasing model size, and they mainly come with a time-consuming deconvolution procedure (Fish et al., 1995). In recent years, kernel-free methods have become more popular in motion deblurring tasks (Tao et al., 2018; Kupyn et al., 2018; 2019; Zamir et al., 2021; Cho et al., 2021; Mao et al., 2023; Nah et al., 2022; Tu et al., 2022; Zamir et al., 2022; Wang et al., 2022; Chen et al., 2022; Tsai et al., 2022; Kong et al., 2023; Li et al., 2023; Kim et al., 2024; 2025), exhibiting strong results without relying on blur kernels. However, they are primarily influenced by the network design (Ronneberger et al., 2015; He et al., 2016; Dosovitskiy et al., 2021; Tolstikhin et al., 2021) and capacity (He et al., 2016; Zagoruyko & Komodakis, 2016; Tan & Le, 2019).

While the kernel-based method can produce better texture-detailed sharp images over the kernel-free method by closely following the blurring process, the kernel-free method is beneficial for effectively managing complex blur patterns. Inspired by this, we propose a hybrid scheme that combines the strengths of kernel-based and kernel-free deblurring methods. Specifically, we decompose a non-uniform deblurring task into a uniform kernel estimation (e.g., managed by our kernel-based method) and error prediction tasks (e.g., managed by our kernel-free method). Our kernel-based model is responsible for estimating a blur kernel, which is used to generate a texture-detailed reference image. Then, our kernel-free model eliminates some errors in the self-generated reference images while preserving the remaining texture details. According to Reference-based Super-Resolution (RefSR) tasks (Zhang et al., 2019; Jiang et al., 2021), they suggest using an additional high-resolution reference image to benefit from meaningful details of the reference image. Namely, leveraging the additional self-generated reference image is a key strategy of our method to recover the realistic details of the deblurred image. In our experiments, we found that the rich-detailed reference image generated from our kernel-based method as shown in Fig. 1 (c) is preferred rather than the artifact-free reference image generated from the recent kernel-free methods as shown in Fig. 1 (b). This is because high texture-detailed reference image can be contributed to restoring the realistic details for deblurring tasks.

To build an efficient kernel-based model for reference training, we pay attention to the Logarithmic Fourier Transform (LogFT) (Brigham & Morrow, 1967; Childers et al., 1977). Our intuition is that the logarithmic fourier space allows for simplifying the blur-sharp relation from complicated deconvolution to subtraction so that learning uniform kernels becomes easier that we can use a light-weight kernel-based model. Furthermore, equipped with a texture-detailed reference image, we can achieve a scaled-down version of our kernel-free model, whose performance is still comparable to that of state-of-the-art methods (Tu et al., 2022; Wang et al., 2022; Chen et al., 2022; Tsai et al., 2022) as shown in Table 1. Extensive experiments have been conducted to demonstrate the superiority of the proposed method under various datasets such as RealBlur (Rim et al., 2020), GoPro (Nah et al., 2017), and RSBlur (Rim et al., 2022) datasets.

## 2 Related Works

**Kernel-based methods.** Kernel-based methods (Schuler et al., 2015; Chakrabarti, 2016; Sun et al., 2015) estimate per-pixel kernels and subsequently apply existing non-blind deblurring methods (Fish et al., 1995; Krishnan & Fergus, 2009) to produce sharp images. To simplify blur kernel estimation, a motion flow estimation method (Gong et al., 2017) is explored to generate motion flow maps. Similarly, MotionETR (Zhang et al., 2021) is developed to estimate motion offsets, which are subsequently used to reconstruct sharp images.

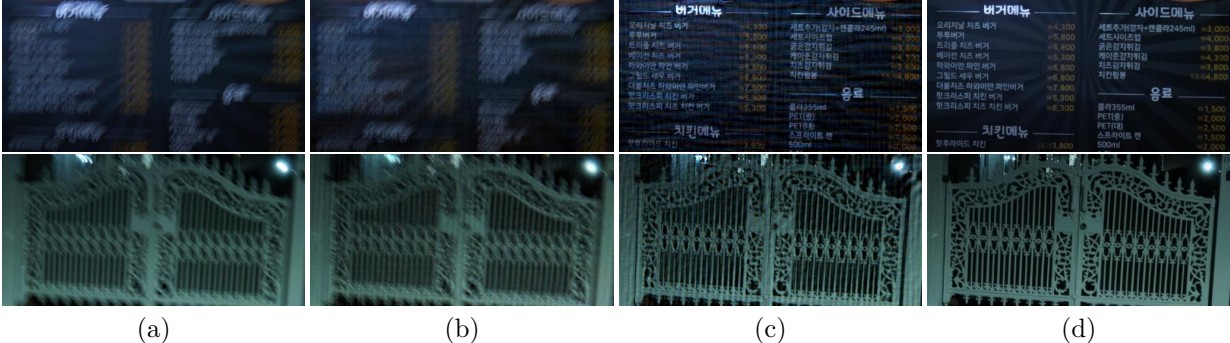

|  (a)  |  (b)  |  (c)  |  (d)  |

Figure 2: Visual results on the proposed method: (a) Blur image (input), (b) Estimated blur image by (6), (c) Estimated reference image by (5), and (d) Final deblurred image by (8). Our kernel-based model is trained with $\lambda = 0.01$.

J-MKPD (Carbajal et al., 2023) proposes an efficient solution by introducing a set of pixel-shared motion bases along with pixel-wise mixing coefficients to accurately estimate per-pixel kernels. Most kernel-based methods struggle to use them in real-world scenarios. This is because they are mainly trained with synthetic data to learn blur kernels or motions. On the other hand, our kernel-based method not only learns blur kernels from realistic blur-sharp pair data (Rim et al., 2020; 2022), but also plays a role to provide a reference image to our kernel-free method for better motion deblurring.

**Kernel-free methods.** The kernel-free methods (Nah et al., 2017; Cho et al., 2021; Chen et al., 2022; Kong et al., 2023; Cui et al., 2024) are based on image-to-image translation tasks, mapping blur images to sharp ones in realistic blur datasets (Rim et al., 2020; 2022). Notably, earlier works (Tao et al., 2018; Nah et al., 2017) introduce a coarse-to-fine strategy that progressively reconstructs sharp images, starting from low-resolution to high-resolution images. In order to implement this strategy more efficiently, some literature (Zamir et al., 2021; Cho et al., 2021; Mao et al., 2023) employs multi-input and multi-output U-Net architectures (Ronneberger et al., 2015). Transformer-based deblurring techniques (Zamir et al., 2022; Wang et al., 2022; Tsai et al., 2022; Kong et al., 2023) are also developed, aiming to enhance image detail recovery by addressing the quadratic complexity of self-attention mechanisms in innovative ways. Additionally, several studies focus on network architecture design to effectively capture both local and global information (Zamir et al., 2021; Tu et al., 2022; Zamir et al., 2022; Li et al., 2023). Recently, prior-based methods (Li et al., 2022; Fang et al., 2023; Kim et al., 2024) have been investigated to further improve deblurring performance. NAFNet (Chen et al., 2022) introduces an efficient network architecture that utilizes simplified channel attention and simple gates. This scheme achieves state-of-the-art performance on the single image deblurring with small computational costs. Our method is based on this efficient kernel-free method but additionally employs a self-generated reference image, which enables us to further reduce computational costs while preserving state-of-the-art performance.

## 3 Reference-Based Blind Motion Deblurring

In this section, we present a new deblurring approach that leverages both kernel-based and kernel-free techniques. We begin by discussing a new approach to handling non-uniform deblurring tasks in Section 3.1. Next, we introduce a logarithmic fourier kernel estimator that yields a reference image in Section 3.2. Section 3.3 covers a reference-based kernel-free method, and finally, we present our hybrid method with implementation details and discussions in Section 3.4.

### 3.1 A new approach to handling non-uniform deblurring

The usage of additional reference images, which aid in restoring the texture details of the image, is a key component of our deblurring scheme. In essence, the image deblurring approaches (Zamir et al., 2021; Cho et al., 2021; Chen et al., 2022) aim to primarily recover the texture details of the image only from a blur image. On the other hand, our hybrid scheme focuses on recovering the texture details of the image from

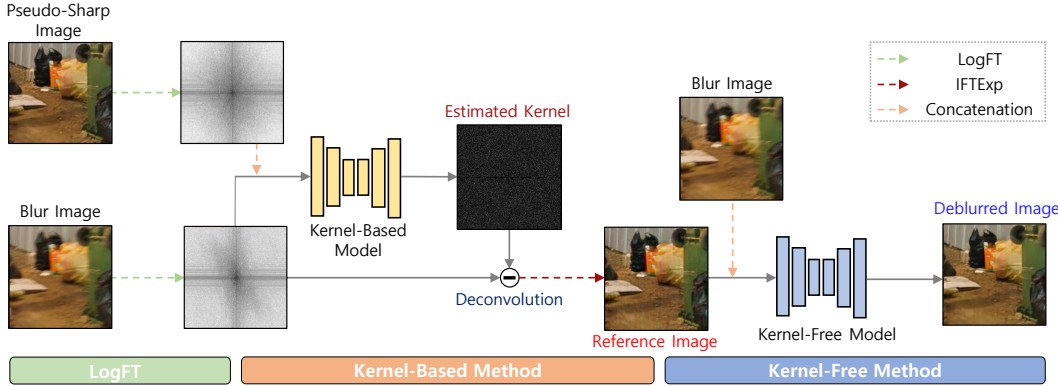

Figure 3: Network architecture overview. Our network architecture consists of two components: kernel-based model and kernel-free model. We use U-Net (Ronneberger et al., 2015) as our kernel-based model. NAFNet (Chen et al., 2022) is adopted for our kernel-free model. LogFT indicates the logarithmic fourier transform and IFTExp means the inverse logarithmic fourier transform.

both a blur and a self-generated reference image. To this end, we first investigate a kernel-based method that follows a blurring process to generate a rich-detailed reference image inevitably with some artifacts. Then, our kernel-free model concentrates on getting rid of the artifacts in such reference images while preserving the remaining texture details.

In the case of uniform deblurring, given a pair of blur image $y$ and uniform blur kernel $k$, the sharp image $x$ is reconstructed by using the following deconvolution process

$$x = y \odot k, \tag{1}$$

where $\odot$ is a deconvolution operation. To extend it to non-uniform deblurring, let the total per-pixel error $e = e_k + e_m$ represent the kernel type error $e_k$ (residual blurs) when the motion blur is not uniform, i.e., non-uniform blur, and the kernel model error $e_m$ (artifacts) when the estimated kernel is not accurate. Then, we approximate a non-uniform deblurred image $\tilde{x}$,

$$
\begin{aligned}
\tilde{x} &= (y \odot k) + e_k = (y \odot (\tilde{k} + \Delta k)) + e_k \\
&= y \odot \tilde{k} + y \odot \Delta k + e_k = y \odot \tilde{k} + e_m + e_k \\
&= y \odot \tilde{k} + e = \tilde{x}_r + e,
\end{aligned}
\tag{2}
$$

where $\tilde{x}_r$ is the reference image and $\Delta k = k - \tilde{k}$ is the difference between real and estimated kernels. Since the kernel model error $e_m$ mathematically contains the blur image in (2), we consider a kernel-free model $f_\theta$ that takes two inputs, e.g., reference image $\tilde{x}_r$ and blur image $y$, to handle two types of errors, e.g. $e = f_\theta(\tilde{x}_r, y)$. According to (2), the non-uniform deblurring problem (i.e., difficult task) is decomposed into uniform kernel estimation (reference estimation) and error prediction, i.e., two simpler tasks. Namely, the non-uniform deblurring can be viewed as reference-based deblurring with a self-generated reference image. To simplify the deconvolution process, we first consider a Discrete Fourier Transform (DFT) of a sample image $y$ as follows:

$$Y(\omega) = \sum_{n=0}^{N-1} y(n) e^{-j\frac{2\pi}{N}\omega n}, \tag{3}$$

where $y(n)$ is a real-valued number at a pixel $n$ and $Y(\omega)$ denotes a complex-valued number at a frequency $\omega$. We remark that discrete fourier transform $F$ naturally converts deconvolution into division, i.e., $y \odot k \leftrightarrows F(y)/F(k)$ (Brigham & Morrow, 1967). To further simplify the relationship between two fourier samples, we introduce a logarithmic operation in the fourier domain, leading to the logarithmic fourier transform $F_L$ of the sample images, and then we obtain the relation $y \odot k \leftrightarrows F_L(y) - F_L(k)$ (Childers et al., 1977).

Let $X^L = F_L(x)$ and $Y^L = F_L(y)$ from a dataset $\mathcal{D} = \{(x, y)\}$ be sharp and blur samples by the logarithmic fourier operator $F_L$. Let $K^L = F_L(k)$ be a blur kernel in the logarithmic fourier space. We are

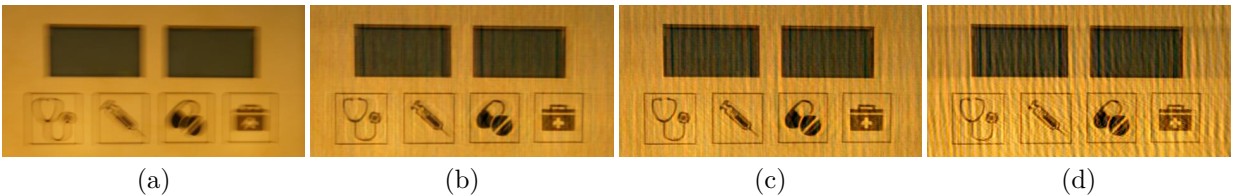

(a)         (b)         (c)         (d)

Figure 4: Reference comparison results: (a) Blur image (input), (b) Reference image with $\lambda = 1.0$, (c) Reference image with $\lambda = 0.01$, and (d) Reference image with $\lambda = 0.001$.

interested in estimating kernel $\tilde{K}^L$ modeled by a kernel-based model $g_\psi$ that maps from $(X^L, Y^L)$ to $\tilde{K}$: we take two inputs, e.g., blur $Y^L$ and sharp image $X^L$ to facilitate estimating accurate kernels as if some literature considers image prior terms (Krishnan et al., 2011) for similar reasons. As the ground-truth sharp image is only valid in the training phase, we instead use the pseudo-sharp image by an off-the-shelf image deblurring model in the evaluation phase. Then, with the estimated kernel $\tilde{K}^L$, the reference image $\tilde{x}_r$ is generated by using simple subtraction (e.g., deconvolution) and inverse logarithmic fourier transform $F_L^{-1}$, i.e., $\tilde{x}_r = F_L^{-1}(Y^L - \tilde{K}^L)$, as shown in Fig. 2 (c). Finally, we obtain the deblurred image $\tilde{x} = \tilde{x}_r + e$ where $\tilde{x}_r$ is the reference image estimated from our kernel-based model $g_\psi$ and $e$ is the total per-pixel error obtained by our kernel-free model $f_\theta$. The whole network architecture is illustrated in Fig. 3.

### 3.2 Logarithmic fourier kernel estimation

In this section, two key schemes are presented to reduce the nature of ill-posedness (make it more accurate) and train a kernel estimator $g_\psi$ under a light-weight network architecture (make it more efficient): (1) we utilize the logarithmic fourier space to learn a blur kernel more easily (Kim et al., 2024) , and (2) we exploit an additional pseudo-sharp image to predict a more accurate kernel. The logarithmic fourier space facilitates learning the blur kernel more easily by simplifying the relationship between blur and sharp samples. Therefore, a light-weight network architecture may be sufficient to accommodate the simplified relation. Furthermore, since non-uniform kernel estimation may require a large network capacity as in (Gong et al., 2017; Zhang et al., 2021; Carbajal et al., 2023), we approximate a non-uniform kernel estimation to a uniform kernel estimation combined with the per-pixel error prediction as discussed in Section 3.1. Specifically, we estimate a uniform kernel under a small-scale network architecture. Then, the per-pixel errors caused by the artifacts and residual blurs are handled in our kernel-free method as explained in Section 3.3. Finally, using an additional pseudo-sharp image restricts the number of feasible kernel solutions, thereby making it simpler to estimate more accurate kernels (i.e., resulting in better references).

To sum up, we aim to train a uniform kernel estimation model $g_\psi$ using the logarithmic fourier pair of input samples, e.g., ground-truth sharp sample [1] $X^L$ and blur sample $Y^L$. Our kernel-based model $g_\psi$ outputs a logarithmic fourier uniform kernel that is expressed as

$$\tilde{K}^L = g_\psi(X^L, Y^L). \tag{4}$$

With the estimated kernel $\tilde{K}^L$, we can easily obtain a self-generated reference image as shown in Fig. 2 (c) by

$$\tilde{x}_r = F_L^{-1}(Y^L - g_\psi(X^L, Y^L)). \tag{5}$$

On the other hand, by taking the simple addition in the logarithmic fourier space, we have

$$\tilde{y} = F_L^{-1}(X^L + g_\psi(X^L, Y^L)), \tag{6}$$

where $\tilde{y}$ is an estimated blur image as shown in Fig. 2 (b). Finally, we suggest minimizing the following loss that coincides with the blurring and deblurring procedures to estimate the accurate kernels:

$$L_{\texttt{kernel}}(\psi; \mathcal{D}) = \frac{1}{|\mathcal{D}|} \sum_{(x,y) \in \mathcal{D}} d(\tilde{y}, y) + \lambda d(\tilde{x}_r, x), \tag{7}$$

---

[1]We observe that learning our kernel-based model with ground-truth sharp images results in better performance than with pseudo-sharp images.

where $d$ is some distance or divergence in the image domain, e.g., the PSNR loss (Chen et al., 2022) and $\lambda$ is a hyperparameter to control the sharpness of the reference image.

Our reference image generation jointly optimizes deblurring and blurring objectives using a shared blur kernel. When emphasizing the deblurring objective (i.e., $\lambda = 1.0$), the reference image becomes artifact-free but lacks texture details, as it tends to prioritize high PSNR performance for the reference image, as shown in Fig. 4 (b). In contrast, prioritizing the blurring objective (i.e., $\lambda = 0.01$) enables more accurate kernel estimation, as the blurring task is significantly easier for the model to learn shared blur kernels than the deblurring task. As a result, this strategy facilitates recovering texture details in the reference image, but introduces some artifacts due to the low contribution of the deblurring loss term $d(\tilde{x}, x)$, as shown in Fig. 4 (c). Further increasing the emphasis on blurring task, e.g., $\lambda = 0.001$, as shown in Fig. 4 (d), may amplify texture recovery but introduces severe artifacts, ultimately degrading the final deblurring performance compared to the case with $\lambda = 0.01$. Further analysis will be explored in Section B.4 of the Appendix.

### 3.3 Kernel-free method with a reference image

To obtain better performance, it is common to scale up a network (He et al., 2016; Zagoruyko & Komodakis, 2016; Tan & Le, 2019) or use texture-detailed reference images (Zhang et al., 2019; Jiang et al., 2021). This section introduces our kernel-free method that exploits such texture-detailed reference images. Our kernel-free method eliminates the per-pixel artifacts (kernel model error $e_m$) and residual blurs (kernel type error $e_k$) using blur and self-generated reference images, as discussed in Section 3.1. Note that our kernel-free method can eliminate the per-pixel residual blurs that are still present in the reference images, especially when dealing with non-uniform blur types.

Given two input images, i.e., reference image $\tilde{x}_r$ and blur image $y$, the deblurred image $\tilde{x}_\theta$ is estimated via our kernel-free model $f_\theta$ as follows:

$$\tilde{x}_\theta = \tilde{x}_r + f_\theta(\tilde{x}_r, y), \tag{8}$$

where $f_\theta(\tilde{x}_r, y)$ is the estimated per-pixel error $\tilde{e}$, $\tilde{x}_\theta$ denotes the final deblurred image as shown in Fig. 2 (d) and $\tilde{x}_r$ is the reference image defined by (5). Then, we derive our deblur loss that minimizes the distance between the real sharp and deblurred images by

$$L_{\mathtt{deblur}}(\theta; \mathcal{D}) = \frac{1}{|\mathcal{D}|} \sum_{(x,y) \in \mathcal{D}} d(\tilde{x}_\theta, x), \tag{9}$$

where $d$ is some distance or divergence in the image domain, e.g., the PSNR loss (Chen et al., 2022).

We highlight that $\ell_1$, $\ell_2$, and PSNR losses are basically specialized to remove noises or artifacts rather than restore texture details. The proposed method is more relevant to use those losses since our kernel-free method is designed for removing artifacts of the texture-detailed reference image.

### 3.4 Comprehensive discussion on our hybrid deblurring framework

**Reference-based blind motion deblurring.** In this section, we present the implementation details of our hybrid method. Basically, our hybrid model consists of two models: kernel-based and kernel-free models. We adopt NAFNet (Chen et al., 2022) as our kernel-free model. We use U-Net (Ronneberger et al., 2015) as our kernel-based model[2]. Since both models take two input images, we simply concatenate them toward the channel dimension to use the original network architectures. In the first stage, we train our kernel-based model $g_\psi$ with the blur and ground-truth sharp images by minimizing (7). In the second stage, we freeze our kernel-based model $g_\psi$ and optimize our kernel-free model $f_\theta$ with the reference $\tilde{x}_r$ (from our kernel-based model) and blur images $y$ by using (9). To generate reference images, the ground-truth sharp images are used in the first stage whereas the pseudo-sharp image is used in the second stage: a pseudo-sharp image is obtained by an off-the-shelf image deblurring model, e.g., NAFNet.

---

[2]The kernel-based model using U-Net achieves comparable performance to NAFNet. However, the training time of NAFNet is longer due to its special modules. Therefore, we choose U-Net for the kernel-based model.

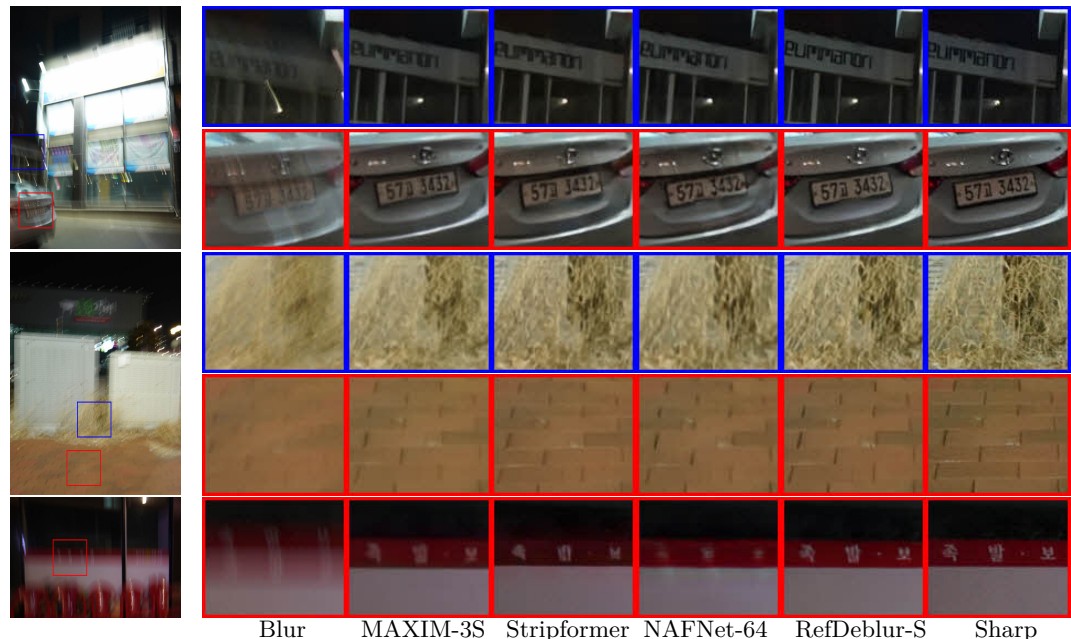

Figure 5: Visual comparison results on RealBlur-J (Rim et al., 2020) dataset. We compare our method with state-of-the-art deblurring methods (Cho et al., 2021; Mao et al., 2023; Tu et al., 2022; Chen et al., 2022; Tsai et al., 2022). The proposed method shows more texture-detailed, artifact-free, and sharper aspects in the deblurred images.

**Discussion on joint training.** We further discuss how our hybrid framework behaves under an end-to-end training regime. While an end-to-end training strategy, which involves jointly training both kernel-based and kernel-free models, is an intuitive and common approach, we found that its performance (33.15 dB) lags behind that of the two-stage learning strategy (33.38 dB). We believe that the proposed step-by-step training leads to more stable convergence and enables each component model (e.g., kernel-based or kernel-free model) to be optimally trained, ultimately resulting in better performance. Nevertheless, our hybrid framework trained jointly (33.15 dB) still outperforms the baseline kernel-free model (32.66 dB) under similar computational costs, highlighting the benefits of the hybrid approach itself.

**Blur kernel constraints.** We do not impose strict physical constraints (e.g., non-negativity and sum-to-one) on the blur kernel, as they may lead to training difficulty. Instead, we apply a [0,1] clamping operation to the generated reference image. This serves two key purposes: (1) ensuring valid pixel ranges for stable input to the subsequent kernel-free model, and (2) implicitly regularizing the kernel-based model to encourage physically-valid kernel estimation without imposing strict physical constraints.

## 4 Experiments

### 4.1 Experimental setup

**Dataset and evaluation metrics.** We use GoPro (Nah et al., 2017), RealBlur (Rim et al., 2020), and RSBlur (Rim et al., 2022) for training and test sets. GoPro contains 2,103 and 1,111 blur-sharp image pairs for training and test sets, respectively. GoPro has been widely used for motion deblurring tasks, but it is a synthetic dataset where the blur image is generated by averaging sharp video frames captured by a high-speed camera. On the other hand, RealBlur and RSBlur are realistic motion blur datasets where they capture blur and sharp images in the same scene by using the beam splitter. RealBlur consists of RealBlur-J (sRGB domain) and RealBlur-R (RAW domain). Each RealBlur type comprises 3,758 and 980 image pairs for training and test sets, respectively. RSBlur dataset contains 8,878 and 3,360 blur-sharp image pairs for training and test sets, respectively. To verify the performance of the proposed method, we measure distortion

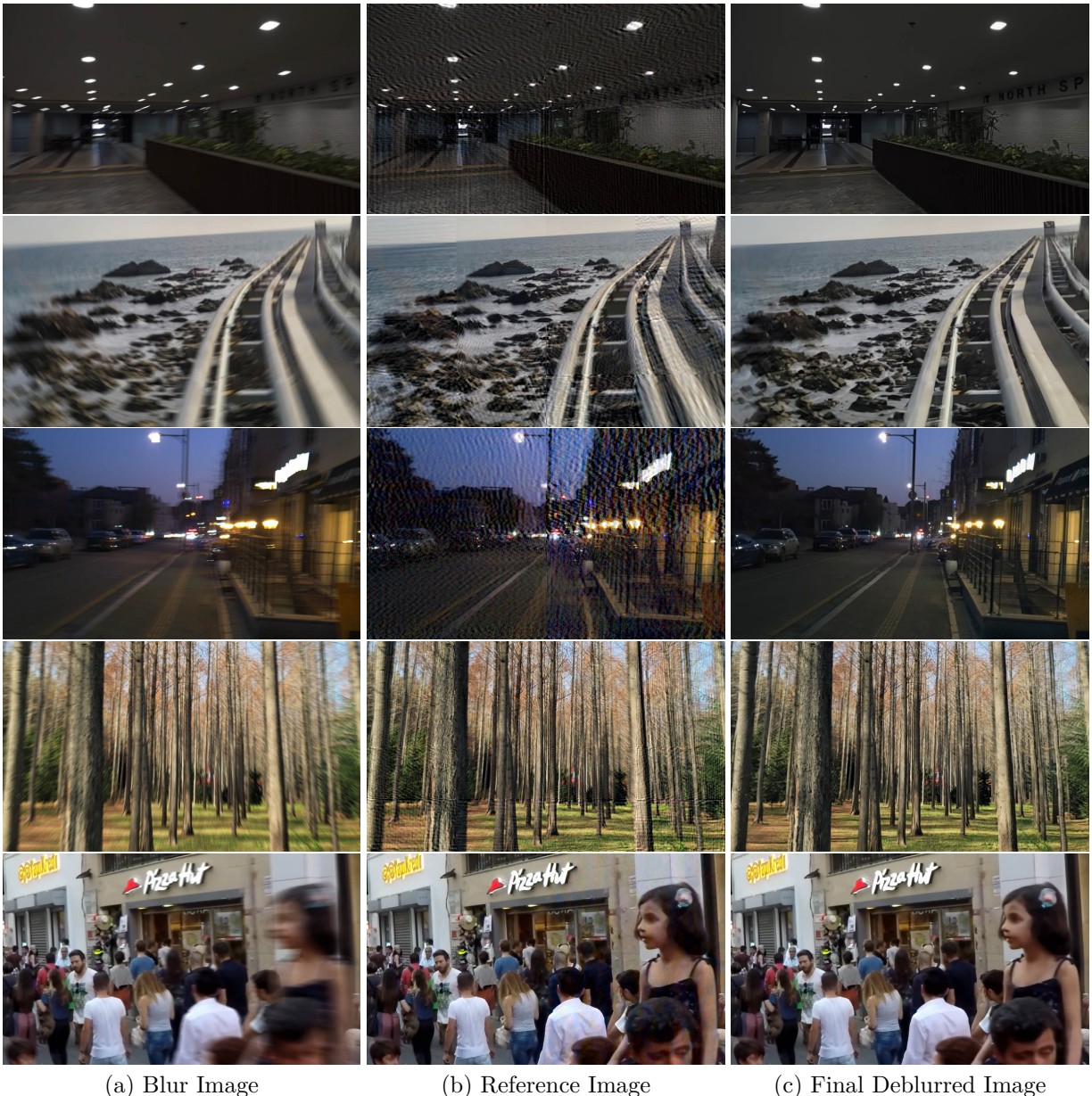

(a) Blur Image (b) Reference Image (c) Final Deblurred Image

Figure 6: Visual results of blur, reference and final deblurred images. Starting from the top row, in order, saturation blur / in-plane rotation blur (e.g., z-axis rotation) / out-of-plane rotation blur (e.g., y-axis rotation) / forward motion blur (e.g., z-axis translation) / object motion blur. The reference images are generated by our kernel-based model trained with $\lambda = 0.01$.

metrics, e.g., Peak Signal to Noise Ration (PSNR) and Structural SIMilarity (SSIM) Wang et al. (2004), and a perceptual metric, e.g., Learned Perceptual Image Patch Similarity (LPIPS) Zhang et al. (2018). To measure model efficiency, we compute the number of network parameters and Multiply–ACcumulate operations (MACs) based on the image size of $256 \times 256$.

**Network architecture variants.** The default number of blocks of NAFNet (Chen et al., 2022) is 36. NAFNet with 16 widths, 32 widths, and 64 widths are referred to as NAFNet-16, NAFNet-32, and NAFNet-64, respectively. Similarly, we refer to our method with 16 widths as RefDeblur-16. To be consistent with the computational cost of NAFNet, we make RefDeblur-T (Tiny) and RefDeblur-S (Small). To cope with large-scale models such as Restormer (Zamir et al., 2022) and MAXIM (Tu et al., 2022), we further increase

Table 1: The effect of the proposed method compared against NAFNet (Chen et al., 2022). The proposed method yields comparable performance with almost a quarter of the computational costs. Under similar computational costs, our method outperforms NAFNet with respect to several scales of network architectures.

| Methods | # Params | GMACs | RealBlur-J | |
|---|---|---|---|---|
| | | | PSNR ↑ | SSIM ↑ |
| NAFNet-16 (Chen et al., 2022) | 4.00 | 4.23 | 31.64 | 0.912 |
| RefDeblur-16 (ours) | 8.39 | 9.65 | 32.37 | 0.921 |
| NAFNet-32 (Chen et al., 2022) | 16.00 | 16.25 | 32.17 | 0.920 |
| RefDeblur-T (ours) | 12.40 | 16.33 | 32.70 | 0.927 |
| NAFNet-64 (Chen et al., 2022) | 63.50 | 63.64 | 32.66 | 0.929 |
| RefDeblur-S (ours) | 54.20 | 62.61 | 33.38 | 0.938 |
| RefDeblur-B (ours) | 123.50 | 136.88 | **33.81** | **0.941** |

the model sizes, which is referred to as RefDeblur-B (Big). Our network variants are described in details in Section A of Appendix.

**Implementation details.** We train with RealBlur (Rim et al., 2020), GoPro (Nah et al., 2017), and RSBlur (Rim et al., 2022) randomly cropped by $256 \times 256$. We train RefDeblur-16 up to $1,000$ epochs (batch size 16) for RealBlur, our RefDeblur variants (T / S / B) up to $2,000$ epochs (batch size 32) for RealBlur, our RefDeblur variants (S / B) up to $1,000$ epochs (batch size 32) for RSBlur and our RefDeblur-B up to $12,000$ epochs (batch size 64) for GoPro. Our kernel-free model is optimized by the AdamW (Loshchilov & Hutter, 2019) algorithm ($\beta_1 = 0.9$, $\beta_2 = 0.9$ and weight decay $1e^{-3}$) with the cosine annealing schedule ($1e^{-3}$ to $1e^{-7}$) gradually reduced for total iterations of each dataset. We optimize our kernel-based model using the Adam algorithm ($\beta_1 = 0.9$, $\beta_2 = 0.9$ and weight decay 0) with the step schedule (initial learning rate $1e^{-4}$) by decreasing the factor of 0.5 every 200 epochs for RealBlur and RSBlur dataset (up to $1,000$ epochs), and every 600 epochs for GoPro dataset (up to $3,000$ epochs). We employ the hyperparameter of ours as $\lambda = 0.01$ and use RealBlur-J for all ablation studies unless otherwise specified.

## 4.2 Reference images for various blur types

We examine that our kernel-based method produces reasonable reference images on several non-uniform blur types such as saturation, in-plane camera rotation (e.g., z-axis rotation), out-of-plane camera rotation (e.g., y-axis rotation), forward motion (e.g., z-axis translation) and object motion blurs. To generate the reference images from the various non-uniform blur types, we collect those blur images from LOL-Blur (Zhou et al., 2022), Heterogeneous Motion Blur (Gong et al., 2017), GoPro (Nah et al., 2017), and real-world blur images captured by Samsung Galaxy Note 20 Ultra. As shown in Fig. 6 (b), under various non-uniform blur types, the reference images represent the sharpened visual results with some entailing artifacts. As discussed in Section 1, our reference images exhibit more ringing artifacts, especially in saturation and low-light (e.g., noisy) regions where the blur kernels become less accurate. Nevertheless, all those artifacts are removed by our kernel-free model as shown in Fig. 6 (c).

## 4.3 Effects of reference-based image deblurring

**Model efficiency.** This experiment aims to show the model efficiency of our method, compared to our baseline, e.g., NAFNet (Chen et al., 2022). We train and evaluate with RealBlur-J dataset (Rim et al., 2020). We measure PSNR and SSIM for model performance while the number of network parameters and MACs are measured for model efficiency. Note that the computational cost of FFT operation is negligibly minor so it is not considered in MACs. As shown in Table 1, the performance of our RefDeblur-T is comparable to that of NAFNet-64, almost with a quarter of the computational costs. Furthermore, our method improves PSNR from 32.66 to 33.38 dB with similar MACs (see the results of NAFNet-64 and RefDeblur-S).

**Quantitative and qualitative results.** We first present the quantitative results on RealBlur-J (Rim et al., 2020). The experimental results are summarized in Table 2. Our RefDeblur-B achieves the best performance, reaching 33.81 dB@PSNR, outperforming previous methods by a significant performance gap.

Table 2: Comparison results on RealBlur-J and R datasets (Rim et al., 2020).

| Methods | GMACs | RealBlur-J | | | RealBlur-R | | |
|---|---|---|---|---|---|---|---|
| | | PSNR↑ | SSIM↑ | LPIPS↓ | PSNR↑ | SSIM↑ | LPIPS↓ |
| MIMO-UNet+ (Cho et al., 2021) | 154.41 | 31.92 | 0.919 | 0.109 | - | - | - |
| MSDI-Net (Li et al., 2022) | 336.43 | 32.35 | 0.923 | - | - | - | - |
| MPRNet (Zamir et al., 2021) | 777.01 | 31.76 | 0.922 | - | 39.31 | 0.972 | - |
| Stripformer (Tsai et al., 2022) | 169.89 | 32.48 | 0.929 | 0.083 | 39.84 | 0.974 | 0.028 |
| MAXIM-3S (Tu et al., 2022) | 169.50 | 32.84 | 0.935 | 0.089 | 39.45 | 0.962 | 0.042 |
| FFTFormer (Kong et al., 2023) | 131.45 | 32.62 | 0.932 | 0.089 | 40.20 | 0.973 | 0.033 |
| GRL-B (Li et al., 2023) | 1285.28 | 32.82 | 0.932 | - | 40.20 | 0.974 | - |
| UFPNet (Fang et al., 2023) | 243.33 | 33.35 | 0.934 | 0.088 | 40.61 | 0.974 | 0.034 |
| FMIMO-UNet (Mao et al., 2023) | 80.21 | 32.65 | 0.931 | 0.100 | 40.01 | 0.972 | 0.044 |
| SegDeblur-L (Kim et al., 2024) | 62.68 | 32.95 | 0.934 | 0.088 | 40.21 | 0.975 | 0.032 |
| NAFNet-64 (Chen et al., 2022) | 63.64 | 32.66 | 0.928 | 0.089 | 39.59 | 0.973 | 0.045 |
| RefDeblur-S (ours) | 62.61 | 33.38 | 0.938 | 0.083 | 40.70 | 0.976 | 0.031 |
| RefDeblur-B (ours) | 136.88 | **33.81** | **0.941** | **0.079** | **41.16** | **0.978** | **0.027** |

Table 3: Comparison results on RSBlur (Rim et al., 2022) dataset. The best results are indicated in bold.

| Methods | GMACs | RSBlur | |
|---|---|---|---|
| | | PSNR ↑ | SSIM ↑ |
| SRN-Deblur (Tao et al., 2018) | 1434.82 | 32.53 | 0.840 |
| MIMO-UNet+ (Cho et al., 2021) | 154.41 | 33.37 | 0.856 |
| MPRNet (Zamir et al., 2021) | 777.01 | 33.61 | 0.861 |
| Restormer (Zamir et al., 2022) | 141.00 | 33.69 | 0.863 |
| Uformer-B (Wang et al., 2022) | 89.50 | 33.98 | 0.866 |
| IRNeXt (Cui et al., 2023) | 114.79 | 34.08 | 0.869 |
| ConvIR-L (Cui et al., 2024) | 129.34 | 34.06 | 0.868 |
| NAFNet-64 (Chen et al., 2022) | 63.64 | 33.65 | 0.862 |
| RefDeblur-S (ours) | 62.61 | 34.01 | 0.867 |
| RefDeblur-B (ours) | 136.88 | **34.24** | **0.871** |

Furthermore, as our method can be interpreted as prior-based deblurring method due to using an additional reference image, we compare it with other prior-based deblurring methods such as MSDI-Net (Li et al., 2022), UFPNet (Fang et al., 2023), and SegDeblur (Kim et al., 2024). We observe that our RefDeblur-S achieves superior deblurring performance compared to the prior-based methods, while requiring small computational cost (62.61 GMACs). Moreover, our proposed two-stage pipeline achieves superior perceptual quality (i.e., better LPIPS scores) compared to the other methods. This suggests that the reference image utilized in our method plays a crucial role in enhancing perceptual quality, thereby leading to better LPIPS scores. As illustrated in Fig. 5, our method shows more texture-detailed and sharper images compared to state-of-the-art deblurring methods. Furthermore, we present qualitative results on real-world scenarios as shown in Fig. 7 of Appendix.

**Results on RAW blur images.** The realistic deblurring process defined by (1) may not hold in the sRGB domain since the acquired RAW blur images (e.g., RealBlur-R (Rim et al., 2020)) are non-linearly transformed to sRGB blur images (e.g., RealBlur-J (Rim et al., 2020)) via Image Signal Processing (ISP) pipelines (Rim et al., 2022). In other words, the RAW domain is a more relevant domain to predict more accurate blur kernels. This implies that our method can produce higher-quality reference images in the RAW domain, leading to further performance improvement. To verify this, we train and evaluate with RealBlur-R. As shown in Table 2, our method significantly improves PSNR performance from 39.59 (NAFNet-64) to 40.70 dB (RefDeblur-S) under similar computational costs. Interestingly, the performance gap in the RAW domain, i.e., RealBlur-R (1.11 dB) between NAFNet-64 and RefDeblur-S is noticeably higher than the gap in the sRGB domain, i.e., RealBlur-J (0.72 dB). Since the blur kernels are more accurately estimated in the RAW domain, the RAW domain is the best demonstration of the effectiveness of our method.

Table 4: Comparison results on GoPro (Nah et al., 2017) dataset. Restoration-based and diffusion-based methods are grouped separately. The best results are indicated in bold.

| Methods | GMACs | GoPro | | |
|---|---|---|---|---|
| | | PSNR ↑ | SSIM ↑ | LPIPS ↓ |
| **Restoration-based Methods** | | | | |
| SRN-DeblurNet (Tao et al., 2018) | 1434.82 | 30.26 | 0.932 | - |
| DeblurGAN-v2 (Kupyn et al., 2019) | 411.34 | 29.55 | 0.934 | - |
| MPRNet (Zamir et al., 2021) | 777.01 | 32.66 | 0.959 | 0.089 |
| MIMO-UNet+ (Cho et al., 2021) | 154.41 | 32.45 | 0.957 | 0.091 |
| Restormer (Zamir et al., 2022) | 141.00 | 32.92 | 0.961 | 0.084 |
| Uformer-B (Wang et al., 2022) | 89.50 | 32.97 | 0.967 | 0.087 |
| Stripformer (Tsai et al., 2022) | 169.89 | 33.08 | 0.962 | 0.078 |
| MAXIM-3S (Tu et al., 2022) | 169.50 | 32.86 | 0.961 | 0.088 |
| IRNeXt (Cui et al., 2023) | 114.79 | 33.16 | 0.962 | 0.084 |
| ConvIR-L (Cui et al., 2024) | 129.34 | 33.28 | 0.963 | 0.083 |
| NAFNet-64 (Chen et al., 2022) | 63.64 | 33.69 | 0.967 | 0.079 |
| NAFNet-96 (Chen et al., 2022) | 141.72 | 33.66 | 0.970 | 0.078 |
| RefDeblur-S (ours) | 62.61 | **33.98** | **0.972** | **0.075** |
| RefDeblur-B (ours) | 136.88 | **34.10** | **0.973** | **0.073** |
| **Diffusion-based Methods** | | | | |
| DvSR-SA (Whang et al., 2022) | - | 33.23 | 0.963 | 0.078 |
| Hi-Diff (Chen et al., 2023) | 142.62 | 33.33 | 0.964 | 0.079 |
| DiffIR (Xia et al., 2023) | 112.75 | 33.20 | 0.963 | 0.078 |

### 4.4 Comparison to kernel-free methods

**Performance on realistic high-resolution blur dataset.** To compare with the recent works under the high-resolution (around $1920 \times 1200$) realistic blur dataset, we train and evaluate with RSBlur (Rim et al., 2022). As shown in Table 3, our RefDeblur-B achieves the best performance over the previous methods. In particular, our RefDeblur-S improves PSNR performance of NAFNet-64 from 33.65 to 34.01 dB under similar computational costs.

**Results on GoPro.** We evaluate our method against recent deblurring methods on GoPro dataset to demonstrate its effectiveness in terms of performance and computational efficiency. As presented in Table 4, our RefDeblur-S outperforms all competing restoration-based methods on the GoPro dataset. It achieves state-of-the-art results across all metrics. Crucially, our RefDeblur-S demonstrates exceptional efficiency, using only 62.61 GMACs, which is substantially lower than most competitive baselines. Furthermore, as shown in Table 4, we compare our method with recent diffusion-based deblurring models such as DvSR-SA (Whang et al., 2022), Hi-Diff (Chen et al., 2023), and DiffIR (Xia et al., 2023). Despite the strong LPIPS performance of diffusion-based methods, our RefDeblur-S attains a better LPIPS (0.075), along with a higher PSNR (33.98 dB) and SSIM (0.972) than all diffusion-based baselines. Unlike diffusion models that require expensive iterative sampling, our RefDeblur-S operates once with lower computational cost. These results demonstrate that our hybrid framework offers a strong performance in both perceptual quality and efficiency, making it well-suited for practical usage.

## 5 Ablation study

**Analyzing the role of each training stage.** In this section, we explore the effectiveness of our two-stage training. As described in Table 5, we begin our ablation study with the pseudo-sharp image generated from NAFNet-16 (Chen et al., 2022), which yields a PSNR of 31.82 dB. When we introduce reference training, i.e., our kernel-based method, to explicitly generate a reference image, we observe a drop in PSNR to 30.16 dB due to the presence of artifacts in the reference image. However, the reference image still contains rich texture information as exemplified in Fig. 1 (c), which is not captured in the pseudo-sharp image, as exemplified in Fig. 1 (b). In the final stage, applying our artifact-free training, i.e., our kernel-free method

Table 5: Ablation study on the effectiveness of our hybrid deblurring framework on the GoPro dataset.

| | | | |
|---|---|---|---|
| Pseudo-Sharp Image | ✓ | ✓ | ✓ |
| Reference Training | | ✓ | ✓ |
| Artifact-Free Training | | | ✓ |
| PSNR (dB) | 31.82 | 30.16 | 33.20 |

Table 6: Comparison results of lightweight models on the GoPro dataset.

| Method | GMACs | PSNR ↑ | SSIM ↑ |
|---|---|---|---|
| NAFNet-32 (Chen et al., 2022) | 16.25 | 32.85 | 0.960 |
| FFTFormer-16 (Kong et al., 2023) | 16.41 | 32.81 | 0.959 |
| RefDeblur-T (ours) | 16.33 | **33.20** | **0.967** |

Table 7: Comparison results between our hybrid deblurring framework and a spatially-variant kernel estimation across different network capacities.

| Method | RefDeblur-T | RefDeblur-S | RefDeblur-B |
|---|---|---|---|
| Spatially-Variant Kernel Estimation | 31.24 | 31.42 | 31.56 |
| Hybrid Deblurring Framework (ours) | **32.70** | **33.38** | **33.81** |

with the reference image, allows the model to eliminate artifacts while retaining fine details. This leads to a significant PSNR improvement to 33.20 dB, notably exceeding the performance of NAFNet-32 (32.85 dB) under the same computational cost. This result clearly demonstrates the effectiveness of our two-stage design, where leveraging a high-detail but flawed reference image leads to better final deblurring results when performing artifact-free training.

**Efficient deblurring training.** To validate the effectiveness of our reference-based design in resource-constrained settings, we evaluate RefDeblur-T (16 GMACs) on the GoPro dataset (Nah et al., 2017) and compare it against other lightweight baselines such as NAFNet-32 (Chen et al., 2022) and FFTFormer-16 (Kong et al., 2023). As shown in Table 6, our method achieves approximately 0.4 dB higher PSNR than NAFNet-32 and FFTFormer-16, demonstrating that the proposed reference-guided deblurring is particularly effective when model capacity is constrained (e.g., small). These results highlight the practical advantage of our framework in efficient deblurring scenarios.

**Spatially-variant kernel estimation vs. our hybrid deblurring framework.** While the kernel-based deblurring, aligned with physical blur modeling, produces texture-rich images, it struggles with accurate spatially-variant kernel estimation due to the training difficulty. Our hybrid framework leverages both: it first uses a kernel-based deblurring, inheriting the capacity to render rich-detailed reference image with some errors. Then, it employs a kernel-free method, specialized in correcting intricate and spatially-variant remaining errors, to effectively remove complex blur degradations. As demonstrated in Table 7, our proposed hybrid deblurring framework demonstrates scalability with increased network capacity and achieves superior performance compared to spatially-variant kernel estimation.

## 6 Conclusions

In this paper, we propose a reference-based motion deblurring scheme that combines kernel-based and kernel-free methods. Our hybrid deblurring scheme combines the strengths of two methods: it initiates with a kernel-based method specialized in restoring high-fidelity details, and then incorporates a kernel-free method specialized in correcting intricate and spatially-variant remaining errors, to effectively removing complex blur degradations. We demonstrate that such hybrid strategy reduces computational costs by 75%, compared to state-of-the-art methods. The proposed method can be extended to other deblurring tasks such as video deblurring (Wang et al., 2019) and defocus deblurring (Abuolaim & Brown, 2020).

## Acknowledgement

This work was supported by Institute for Information & communications Technology Promotion(IITP) grant funded by the Korea government(MSIT) (No.RS-2019-II190075, Artificial Intelligence Graduate School Support Program(KAIST); No.RS-2021-II212068, Artificial Intelligence Innovation Hub; No. RS-2024-00509279, Global AI Frontier Lab).

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

# A   Implementation Details

**Logarithmic operation.** For stable training, we add $10^{-12}$ when performing the complex logarithmic operation. Furthermore, we stabilize our kernel-based model by initializing the model weights to have the normal distribution (mean : 0, standard deviation: 0.02). Hence, the estimated kernel begins by the zero kernel in the logarithmic foruier domain (equivalent to the identity kernel in the spatial domain).

**Network architecture variants.** Our hybrid model consists of two models: kernel-based and kernel-free models. We adopt NAFNet (Chen et al., 2022) as our kernel-free model. We use U-Net (Ronneberger et al., 2015) as our kernel-based model because it empirically shows the consistent performance in various datasets. Also, we consider the pre-trained NAFNet for generating pseudo-sharp images used in our kernel-based model. The default number of blocks of NAFNet (Chen et al., 2022) is 36 which consists of encoder blocks $\{2, 2, 2, 20\}$, middle block $\{2\}$ and decoder blocks $\{2, 2, 2, 20\}$. NAFNet with 16 widths, 32 widths, and 64 widths are referred to as NAFNet-16, NAFNet-32, and NAFNet-64, respectively. Similarly, we refer to our method with 16 widths as RefDeblur-16 as shown in Table 8. To be consistent with the computational cost of NAFNet-32 and 64, we increase the model size of the pseudo-sharp model (NAFNet-16 → NAFNet-16 or NAFNet-32), our kernel-based (UNet-16 → UNet-32) and kernel-free model (NAFNet-16 → NAFNet-16+ or NAFNet-32+) in RefDeblur-16, which are denoted as RefDeblur-T and RefDeblur-S as shown in Table 8. To cope with the large-scale models such as MAXIM (Tu et al., 2022), Restormer (Zamir et al., 2022), and Stripformer (Tsai et al., 2022), we further increase the model size of the pseudo-sharp (NAFNet-32 → NAFNet-64) and kernel-free model (NAFNet-32+ → NAFNet-32++) in RefDeblur-S, which is referred to as RefDeblur-B as shown in Table 8. Here, NAFNet16+ consists of encoder blocks $\{6, 6, 6, 22\}$, middle block $\{6\}$ and decoder blocks $\{6, 6, 6, 6\}$ with 16 widths; NAFNet-32+ consists of encoder blocks $\{9, 9, 9, 25\}$, middle block $\{9\}$ and decoder blocks $\{9, 9, 9, 9\}$ with 32 widths; and NAFNet-32++ consists of encoder blocks $\{16, 16, 16, 32\}$, middle block $\{16\}$ and decoder blocks $\{16, 16, 16, 16\}$ with 32 widths. Note that we use a variant of U-Net that performs four times of downsampling by stride 2 operations. Overall, our models are summarized in Table 8.

Table 8: Our model variants. We present a detailed description of our individual variants such as the sub-models types and the computational costs. The individual GMACs of the sub-models are denoted in brackets.

| Our Model | Model (GMACs) | | | Total GMACs | Total # Params |
|---|---|---|---|---|---|
| | Pseudo-Sharp | Kernel-Based | Kernel-Free | | |
| RefDeblur-16 | NAFNet-16 (4.23) | UNet-16 (1.16) | NAFNet-16 (4.26) | 9.65 | 8.39 |
| RefDeblur-T | NAFNet-16 (4.23) | UNet-32 (3.99) | NAFNet-16+ (8.11) | 16.33 | 12.40 |
| RefDeblur-S | NAFNet-32 (16.25) | UNet-32 (3.99) | NAFNet-32+ (42.37) | 62.61 | 54.20 |
| RefDeblur-B | NAFNet-64 (63.64) | UNet-32 (3.99) | NAFNet-32++ (69.25) | 136.88 | 123.50 |

# B   Additional ablation study

## B.1   Types of reference images

The goal of this experiment is to verify which types of reference images are appropriate. To generate various types of reference images, the pseudo-sharp images trained with PSNR, perceptual (VGG) (Johnson et al., 2016), and VGG + Generative Adversarial Networks (GAN) (Goodfellow et al., 2014) losses are chosen as reference images. Also, other kernel-based methods such as J-MKPD (Carbajal et al., 2023) and MotionETR (Zhang et al., 2021) are considered as reference generators. Note that we consider the VGG and VGG+GAN losses because they are widely studied to perceptually recover texture-detailed images in image restoration tasks (Johnson et al., 2016; Ledig et al., 2017). Given those reference images, we train our kernel-free models with them. The PSNR performance of reference and deblurred images are presented in Table 9.

Although the reference images by VGG and VGG + GAN losses bring in better final deblurring results compared with the result of "No Reference", their performance improvements are relatively small when compared to our method: some texture details from VGG and VGG + GAN may appear in unpleasant visual artifacts rather than realistic details in the image domain (Liang et al., 2022). Furthermore, since the other methods aim for a high PSNR performance of the reference image, it does not help produce texture-detailed reference images. As a reference image, prioritizing the inclusion of high-frequency realistic details even with some artifacts is more crucial for achieving better final deblurring results compared to focusing solely on a higher reference PSNR (compare the results of "RefDeblur-16" with those of "NAFNet-16 with PSNR"). Also, it is proven that the reference image is not always helpful for final deblurring performance (compare the results between "No Reference" and "J-MKPD" and "MotionETR").

Table 9: Ablation study on types of reference images.

| Reference Model | | | Ref. PSNR | Deblur PSNR |
|---|---|---|---|---|
| Type | Loss | GMACs | | |
| NAFNet-16 | PSNR | 4.23 | **31.64** | 31.73 |
| | VGG (Johnson et al., 2016) | 4.23 | 29.56 | 31.76 |
| | VGG+GAN (Goodfellow et al., 2014) | 4.23 | 30.81 | 31.80 |
| RefDeblur-16 | PSNR | 5.39 | 27.89 | **32.37** |
| No Reference | | - | - | 31.58 |
| J-MKPD (Carbajal et al., 2023) | | 56.36 | 28.88 | 31.32 |
| MotionETR (Zhang et al., 2021) | | 55.92 | 28.04 | 31.28 |

## B.2 Effects on pseudo-sharp images

In this section, we analyze how the pseudo-sharp images affect the performance of our kernel-based model. As discussed in Section 3.2, we use the pseudo-sharp image because the paired input can restrict the number of feasible kernel solutions, which results in more accurate kernels and better final deblurring results. We adopt a training strategy in which the kernel-based model is trained with Ground-Truth (GT) sharp images and evaluated using pseudo-sharp images. This strategy is motivated by two primary reasons. Firstly, its final deblurring performance is comparable to training directly with pseudo-sharp images. More importantly, this GT-based training strategy provides crucial extensibility: it allows us to utilize pseudo-sharp images generated by any off-the-shelf deblurring model without requiring retraining of our kernel-based model. To validate this extensibility and demonstrate the impact of pseudo-sharp image quality, we generate reference images using pseudo-sharp images produced by external deblurring models such as NAFNet (Chen et al., 2022), FFTFormer (Kong et al., 2023), and Restormer (Zamir et al., 2022). Then, we use these reference images to train our kernel-free models. As shown in Table 10, these results show that employing higher-quality pseudo-sharp images yields better final deblurring performance. This directly demonstrates that leveraging stronger pseudo-sharp images helps mitigate the potential training-inference gap (e.g., training with GT, but evaluating with pseudo-sharp images), leading to better final deblurring performance.

Table 10: Effect of different pseudo-sharp images on RefDeblur-B.

| Input Types | Pseudo-Sharp Model | | | Final Deblurring Performance | |
|---|---|---|---|---|---|
| | Network | PSNR | LPIPS | PSNR | SSIM |
| Blur + Pseudo-Sharp | NAFNet | 32.66 | 0.089 | **33.81** | **0.941** |
| | FFTFormer | 32.62 | 0.089 | 33.22 | 0.937 |
| | Restormer | 32.32 | 0.108 | 32.88 | 0.933 |

## B.3 Effects on model size

Our hybrid model consists of kernel-based and kernel-free models. To obtain a pseudo-sharp image for our kernel-based model, an additional off-the-shelf pseudo-sharp model, e.g., pre-trained NAFNet (Chen et al., 2022) is considered. Here, we raise a fundamental question: which model size should we increase to improve performance? To address this, we conduct experiments with various model sizes by scaling the size

of pseudo-sharp (PS), kernel-based (KB), and kernel-free (KF) models. The results are shown in Table 11. It is reported that the small-scale kernel-based model is sufficient to provide high performance (compare the performance in the 1st-2nd-3rd row results). Furthermore, we observe that the size of the kernel-free model is the most important ingredient to improve performance. This is witnessed by comparing the 2nd-3rd (KB size increase), 2nd-4th (PS size increase), and 2nd-5th (KF size increase) rows for total MACs and PSNR.

Table 11: Ablation study on model sizes of our method. We make several combinations of pseudo-sharp, kernel-based, and kernel-free models when varying the size of each model. The bolded GMACs indicate the increase compared to the previous row.

| MACs (G) | | | | PSNR | SSIM |
|---|---|---|---|---|---|
| Pseudo-Sharp | Kernel-Based | Kernel-Free | Total | | |
| 16.25 | 1.16 | 16.30 | 33.71 | 32.92 | 0.931 |
| 16.25 | **3.99** | 16.30 | 36.54 | 33.00 | 0.931 |
| 16.25 | **14.69** | 16.30 | 47.24 | 33.00 | 0.931 |
| **63.64** | 3.99 | 16.30 | 83.93 | 33.28 | 0.933 |
| 16.25 | 3.99 | **42.37** | 62.61 | 33.38 | 0.938 |
| 16.25 | 3.99 | **69.25** | 89.49 | 33.60 | 0.940 |

## B.4 Effects on $\lambda$

We investigate which hyperparameter $\lambda$ produces the most suitable reference image. We experiment with $\lambda = \{0.001, 0.01, 0.1, 1.0\}$. Given the reference images varying $\lambda$ from 0.001 to 1.0, the final deblurring results are shown in Table 12. Although low $\lambda$ values, such as 0.01 or 0.1, lead to some artifacts, as shown in Fig. 4 (c), they provide a richer, more detailed reference image, thereby resulting in better deblurring performance. On the other hand, when $\lambda$ is too small, such as 0.001, as shown in Fig. 4 (d), it gives rise to a large amount of artifacts, so that such high dominance of artifacts in the reference image results in degraded performance despite its high-frequency details contained. Therefore, we choose the hyperparameter $\lambda$ as 0.01 in our experiments.

Table 12: Ablation study on hyperparameter $\lambda$. We investigate the hyperparameter $\lambda$ since it is related to the quality of the reference image, which influences the final performance.

| Methods | $\lambda$ | PSNR ↑ | SSIM ↑ |
|---|---|---|---|
| NAFNet-16 | - | 31.58 | 0.912 |
| RefDeblur-16 | 0.001 | 32.14 | 0.919 |
| | 0.01 | **32.37** | **0.921** |
| | 0.1 | 32.24 | 0.918 |
| | 1.0 | 32.20 | 0.916 |

## B.5 Iterative evaluation of the proposed pipeline

We investigate the effect of applying our pipeline iteratively, where the deblurred image generated from the previous iteration is reused as a new pseudo-sharp image for the next. This strategy aims to examine whether repeated refinement of our pipeline can lead to further improvements in deblurring performance. As shown in Table 13, we observe that performance tends to degrade with subsequent iterations. We believe this is because the pseudo-sharp images generated from previous iterations differ in characteristics from those the kernel-free model was originally trained with. This can lead to 'unseen' input for the next iteration, leading to performance degradation. To confirm this, we perform an additional experiment where the kernel-free model is retrained for the second iteration. In this setup, we use the deblurred images from the first iteration as new pseudo-sharp images for the second. Interestingly, as shown in Table 13, this retrained kernel-free model shows improved performance. This result supports our assumption that the initial performance drop is caused by the second kernel-free model encountering out-of-distribution pseudo-sharp images when reused iteratively without retraining.

Table 13: Iterative evaluation of the proposed pipeline on RefDeblur-T. Performance is measured across iterations, with and without retraining the kernel-free model using updated pseudo-sharp images.

| Evaluation Setting | Retraining | PSNR | SSIM |
|---|---|---|---|
| 1st Iteration | | 32.70 | 0.927 |
| 2nd Iteration | | 32.62 | 0.924 |
| 3rd Iteration | | 32.57 | 0.924 |
| 2nd Iteration | ✓ | **32.83** | **0.928** |

### B.6 Spatial vs. Fourier vs. LogFourier

We examine whether the logarithmic fourier space is the most effective input domain in our kernel-based method. To this end, we train our kernel-based models using input samples in spatial, fourier, and logarithmic fourier domains to make reference images which are then used to train our kernel-free models. The final deblurring results are reported in Table 14. The results show that the logarithmic fourier space leads to the best deblurring performance among the domains since the logarithmic fourier domain helps enables to find more accurate blur kernels by simplifying the relationship between sharp and blur images (e.g., deconvolution to subtraction).

Table 14: Ablation study on input domains in our kernel-based method.

| Architecture | Kernel-Based Input Domain | Kernel-Free Model | |
|---|---|---|---|
| | | PSNR | SSIM |
| NAFNet-16 | - | 31.58 | 0.912 |
| RefDeblur-16 | Spatial | 31.74 | 0.913 |
| | Fourier | 32.20 | 0.919 |
| | LogFourier | **32.37** | **0.920** |

## C  Generalization to real-world blur images

We capture real-world blur images with natural hand motions by Samsung Galaxy Note 20 Ultra. We use our *RefDeblur-S* model trained with RealBlur-J (Rim et al., 2020) in order to show the effectiveness of our method under the lowest complexity, compared with the previous methods such as Stripformer (Tsai et al., 2022) and NAFNet-64 (Chen et al., 2022). As shown in Fig. 7, the real-world blur images are well-reconstructed by our method. Meanwhile, the compared methods work well on some blur images but some other blur images are not well-recovered. This shows a better generalization of our method compared with other methods.

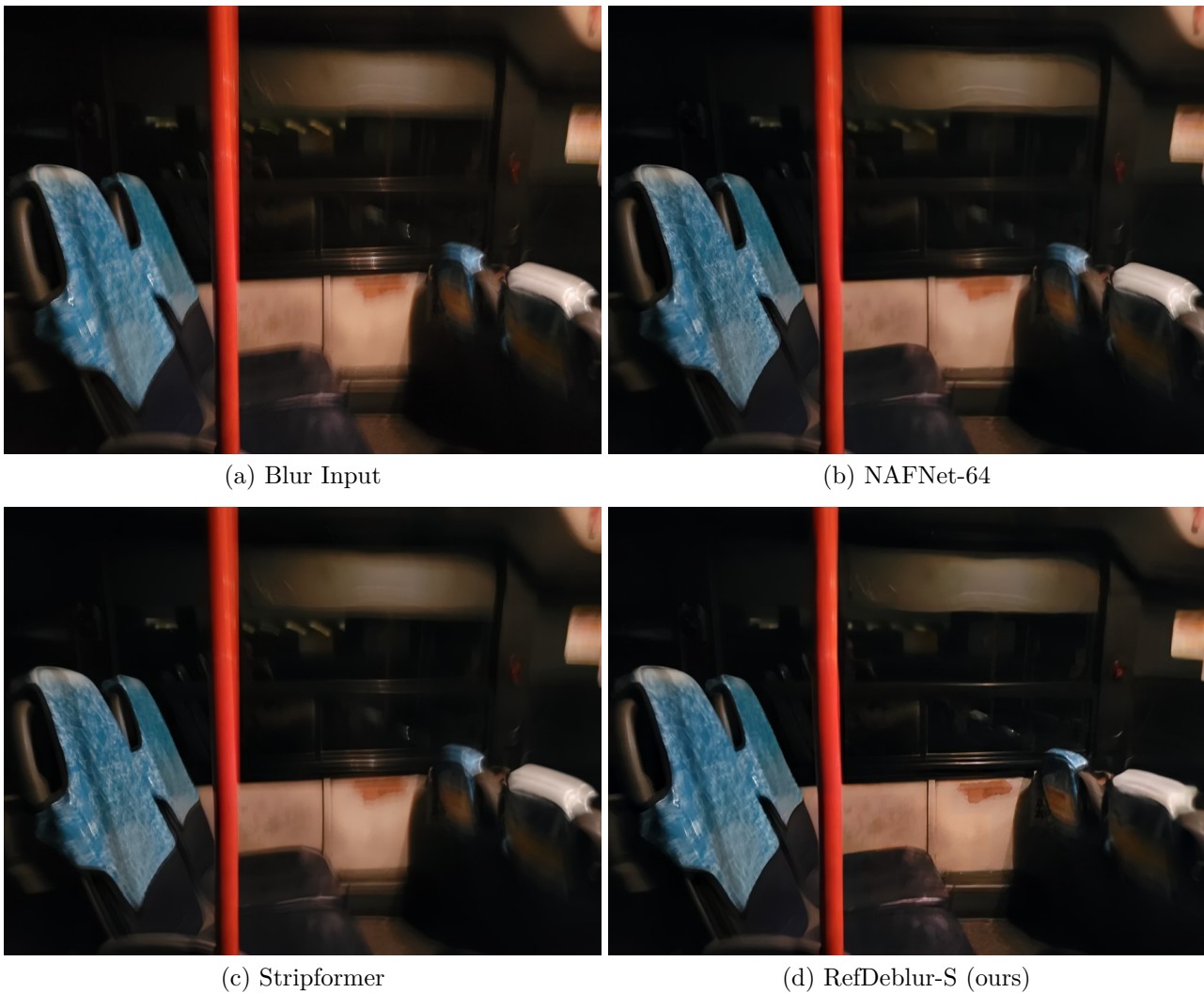

(a) Blur Input

(b) NAFNet-64

(c) Stripformer

(d) RefDeblur-S (ours)

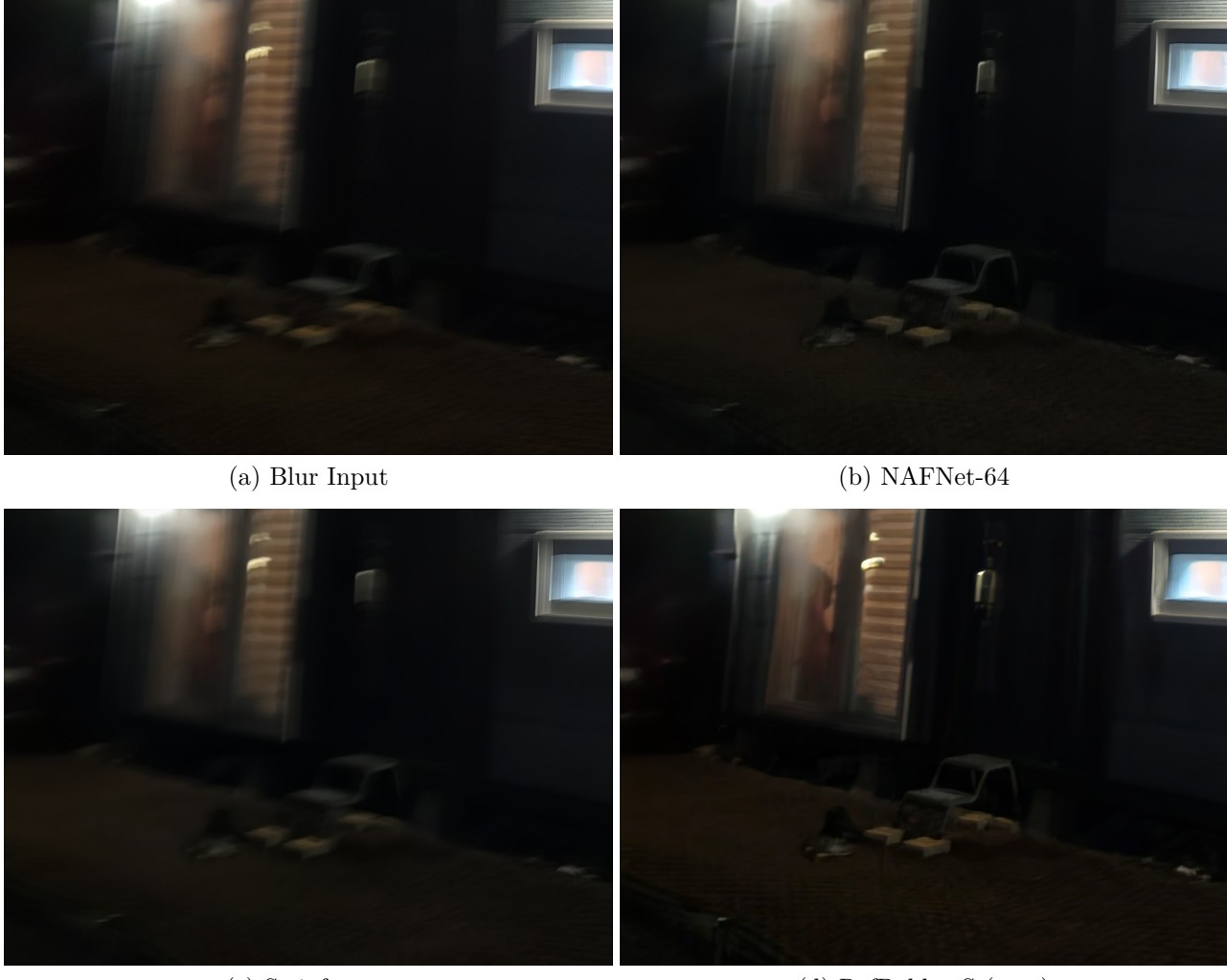

(a) Blur Input

(b) NAFNet-64

(c) Stripformer

(d) RefDeblur-S (ours)

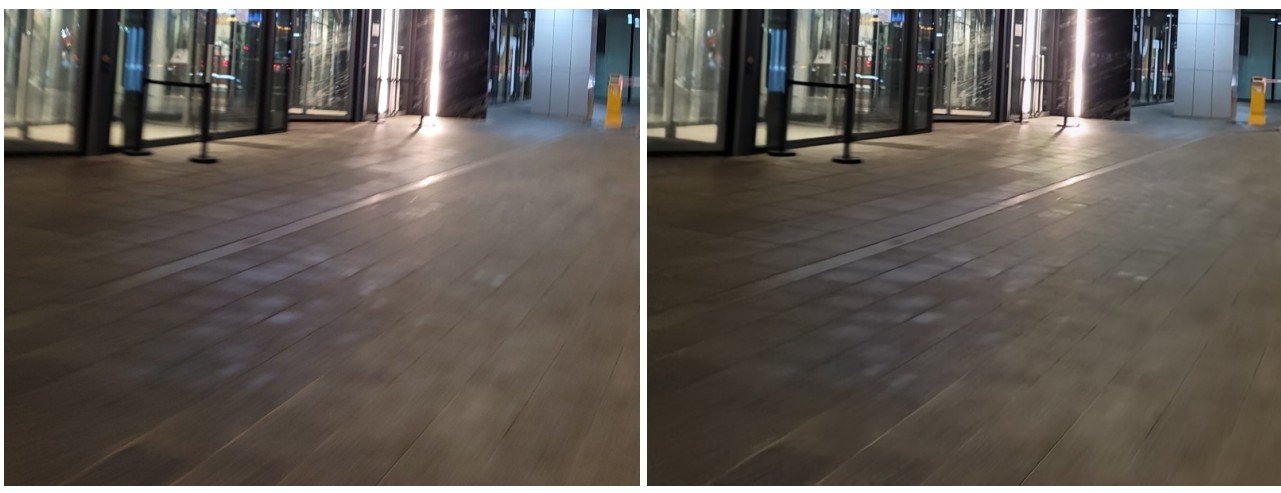

(a) Blur Input                                     (b) NAFNet-64

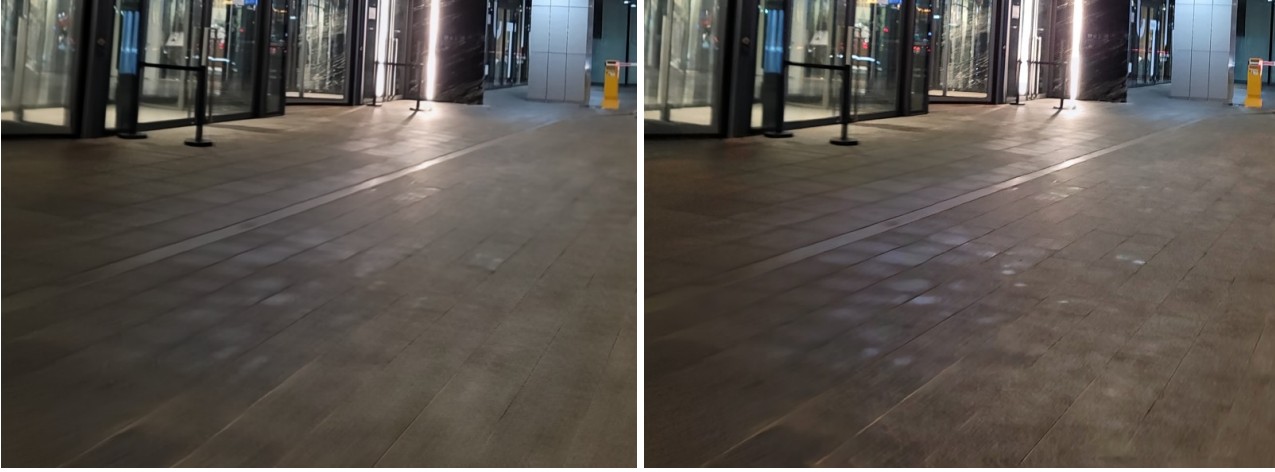

(c) Stripformer                                    (d) RefDeblur-S (ours)

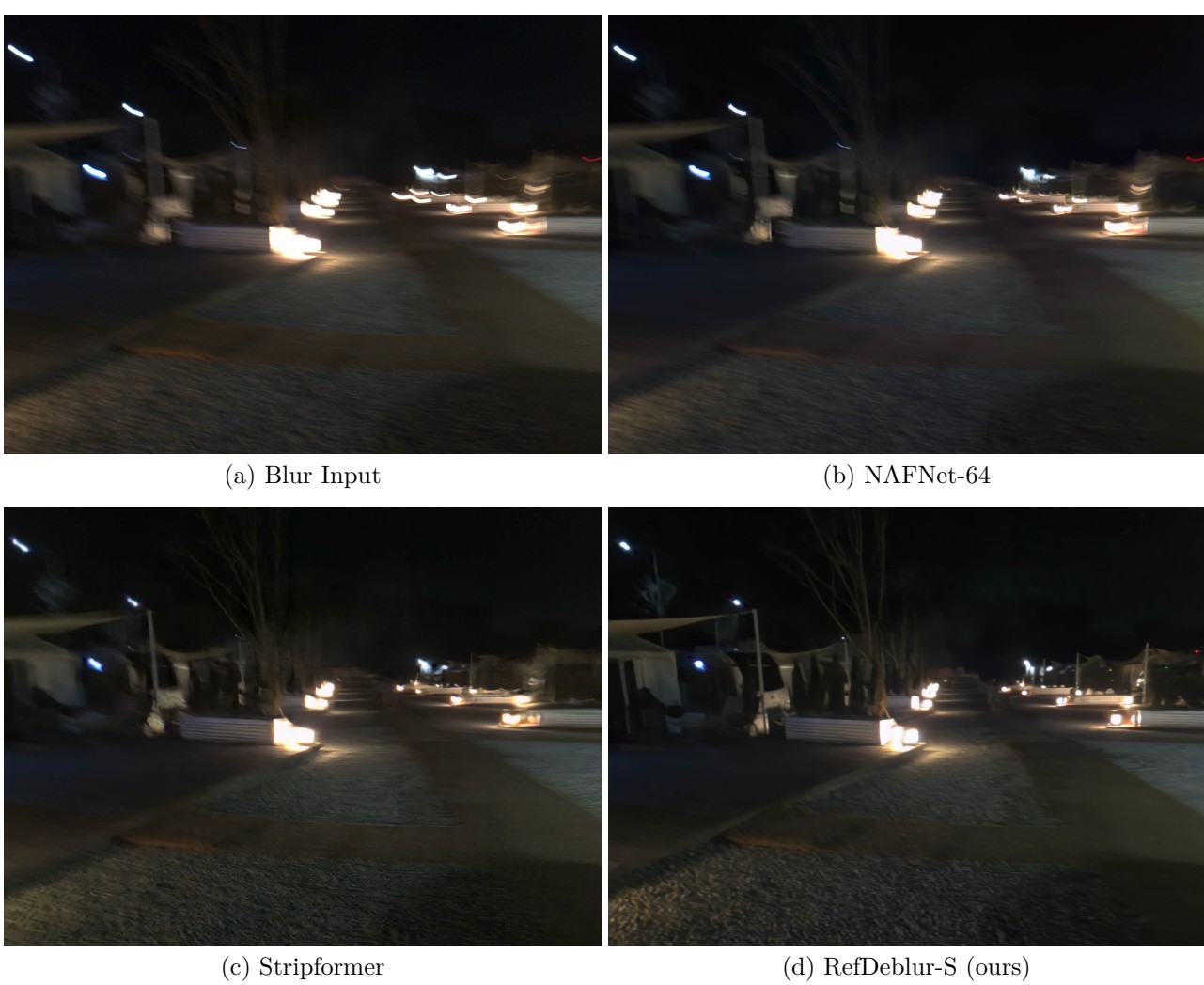

(a) Blur Input

(b) NAFNet-64

(c) Stripformer

(d) RefDeblur-S (ours)

Figure 7: Qualitative results on real-world scenarios.

# D    Additional qualitative results

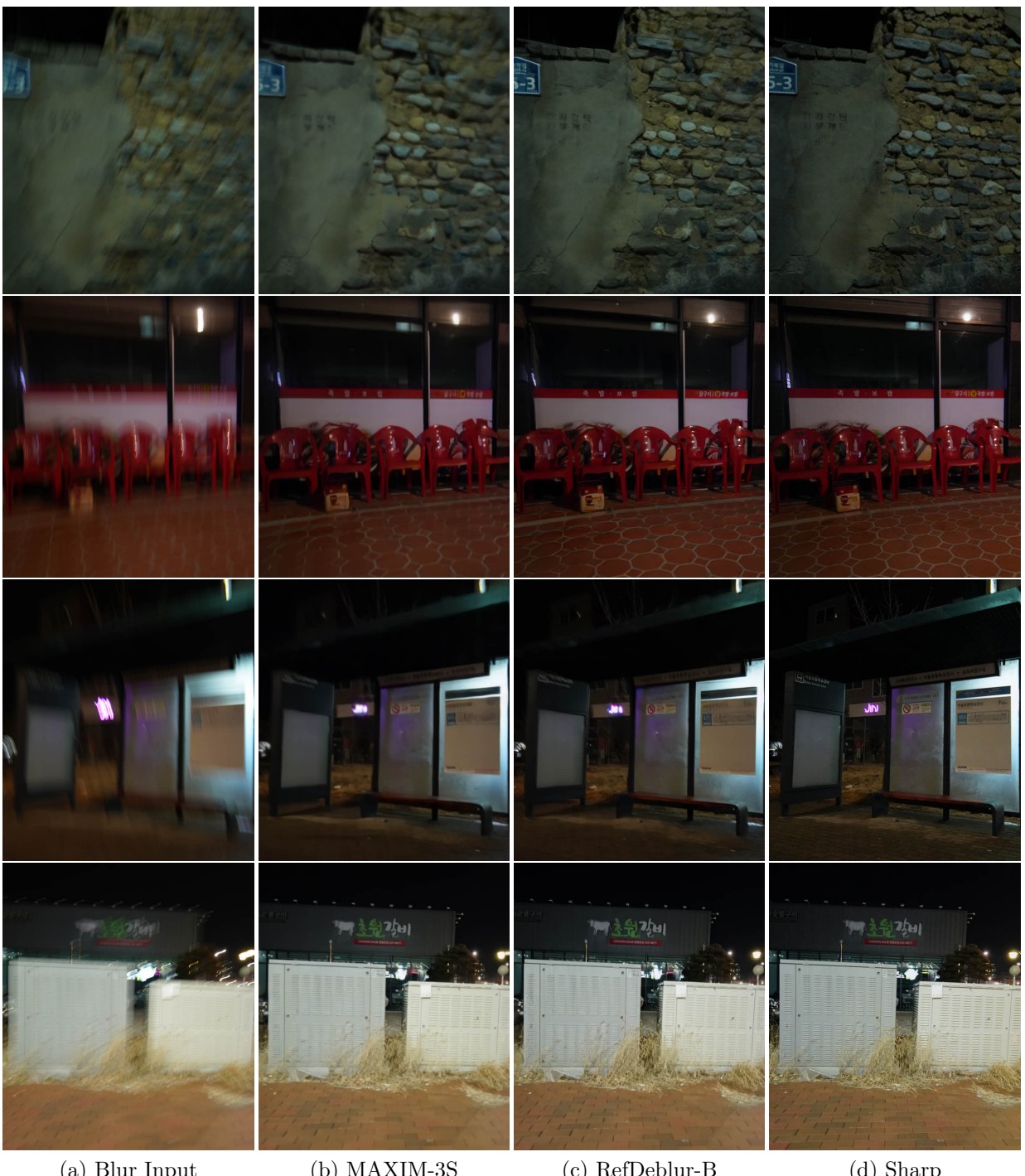

    (a) Blur Input           (b) MAXIM-3S         (c) RefDeblur-B        (d) Sharp

Figure 8: Additional qualitative results on RealBlur (Rim et al., 2020).

