# OpenReview forum: "RefDeblur: Blind Motion Deblurring with Self-Generated Reference Image"
_TMLR — Accepted by TMLR_

### Review · Reviewer_tHgJ · 2025-04-29

**Summary Of Contributions:**

This paper proposes to combine kernel estimation and kernel-free method as a hybrid framework for blind motion deblurring. More specifically, the kernel estimation is used for generating a reference image and kernel-free method refines it by correcting residual errors. The proposed method is evaluated on several datasets to demonstrate its performance.

**Audience:**

Yes

**Broader Impact Concerns:**

No concerns.

**Claims And Evidence:**

Yes

**Requested Changes:**

Please see **Weaknesses**.

**Strengths And Weaknesses:**

Strengths:
1. The proposed combination of kernel-based and kernel-free methods for motion deblurring is interesting.
2. The idea of self-generated reference image also sounds interesting.

Weaknesses:
1. Why do the authors use different networks for the kernel-based and kernel-free models? NAFNet has a similar but stronger architecture than U-Net. Why don't you use it for all stages?
2. In comparison, if the authors' goal is to recover a sharper, more realistic result based on the reference image, it might be good to add the LPIPS metric for the main tables. And I would also suggest comparing with at least one diffusion-based approach.
3. Can you remove the individual learning of kernel estimation by training the whole framework end-to-end with a single final reconstruction loss?
4. In the first paragraph of the introduction, motion blur also occurs due to high-speed movement.
5. For both Figure 1 and Figure 2, it would be nice to have larger, multiple examples to illustrate the effectiveness of the proposed method. In addition, please explain what $\lambda$ is in the caption.

---

> ### Author Response · Authors · 2025-06-07
> **Official Comment by Authors**
>
> Dear Reviewer tHgJ,
>
> We sincerely thank the reviewer for your valuable comments and efforts. The insights have helped us clarify our contributions. We have revised our draft (highlighted in blue) in response to the insightful and valuable comments. We do our best to clarify each of the concerns carefully as follows:
>
> ***
> $$\textbf{(W1) Why using different networks for the kernel-based and kernel-free models?}$$
>
> We appreciate the reviewer’s insightful question. We already experimented with both U-Net and NAFNet architectures for our kernel-based model. Although NAFNet has a more recent and stronger network design, our experiments show that its performance is comparable to U-Net in the final deblurring results. However, the training time of NAFNet in the kernel-based model is significantly longer due to its special modules. Given this result, we chose U-Net for the kernel-based model to ensure more efficient training while maintaining similar performance. We have added this discussion in the revised draft. Please see the footnote of Section 3.4.
>
> ***
> $$\textbf{(W2) Add LPIPS metric and compare it with diffusion-based methods.}$$
>
> Thank you for the valuable suggestions. We have added the LPIPS metric to Tables 2 (RealBlur) and 4 (GoPro) to compare the perceptual quality across methods in the revised draft. Note that we evaluate LPIPS using publicly available pre-trained models. However, for the RSBlur dataset, the pre-trained models are not released, making it infeasible to report LPIPS results for this dataset. As shown in Table 2 and 4, the results show that our proposed two-stage pipeline achieves superior perceptual quality compared to the other methods. This suggests that the reference image utilized in our method plays a crucial role in enhancing perceptual quality, thereby leading to better LPIPS scores. We include this discussion in Section 4.3 (e.g., Quantitative and qualitative results) of the revised draft.
>
> In addition, we have included comparisons with some diffusion-based methods in Table 4 (GoPro) of the revised draft. Despite the strong LPIPS performance of diffusion-based methods, our RefDeblur-S attains a better LPIPS (0.075), along with a higher PSNR (33.98 dB) and SSIM (0.972) than all diffusion-based baselines. Furthermore, unlike diffusion models that require expensive iterative sampling, our RefDeblur-S operates once with lower computational cost. These results demonstrate that our hybrid framework offers a strong performance in both perceptual quality and efficiency, making it well-suited for practical usage. We include this discussion in Section 4.4 (e.g., Results on GoPro) of the revised draft.
>
> ***
> $$\textbf{(W3) Training the whole framework in an end-to-end manner.}$$
>
> Thank you for the constructive suggestion regarding learning strategies. We had indeed considered and experimented with both end-to-end and two-stage learning for our hybrid framework. While an end-to-end training strategy, which involves jointly training both kernel-based and kernel-free models, is an intuitive and common approach, we found that its performance (33.15 dB) lags behind that of the two-stage learning strategy (33.38 dB). We believe that the proposed step-by-step training leads to more stable convergence and enables each component model  (e.g., kernel-based or kernel-free model) to be optimally trained, ultimately resulting in better performance. Nevertheless, our hybrid framework trained jointly (33.15 dB) still outperforms the baseline kernel-free model (32.66 dB) under similar computational costs, highlighting the benefits of the hybrid approach itself. We have updated our draft with this detailed discussion in Section 3.4 (e.g., Discussion on joint training).
>
> ***
> $$\textbf{(W4) Refine the first paragraph of the introduction.}$$
>
> Thank you for the advice. We have clarified the sentence in the introduction section of the revised draft.
>
> ***
> $$\textbf{(W5) Update Figures 1 and 2 with the corresponding λ values.}$$
>
> Thank you for the suggestion. We have updated Figure 1 and 2 with larger and more representative examples and added new Figure 4, to better illustrate the effectiveness of the proposed method in the revised draft. Please refer to Figure 1,2 and 4 of the revised draft.

---

> > ### Comment · Reviewer_tHgJ · 2025-06-24
> > **Official Comment by Reviewer tHgJ**
> >
> > I would like to thank the authors for their efforts in the revision. Most of my concerns are addressed.

---

> > > ### Author Response · Authors · 2025-06-25
> > > **Official Comment by Authors**
> > >
> > > Dear Reviewer tHgJ,
> > >
> > > We are glad to hear that we have addressed most of the reviewer's concerns. The valuable and constructive feedback has helped us clarify our contribution. Thank you once again for the thoughtful review and the opportunity to refine our work. Please feel free to reach out with any additional questions or suggestions.
> > >
> > > Best regards,
> > >
> > > Authors

---

### Review · Reviewer_2gAJ · 2025-05-14

**Summary Of Contributions:**

In this paper, authors propose hybrid framework that decomposes a non-uniform deblurring task into two simpler tasks, i.e., a uniform kernel estimation and error prediction, which are handled by kernel-based and kernel-free methods, respectively. To be specific, the kernel-based sub-module aim to generate a reference image with realistic texture details, and kernel-free sub-module concentrate on refining the reference image by correcting residual errors with preserving texture details. Besides, logarithmic fourier space and regime are introduced into the methods. In the experimental part, authors claim that their proposed method achieves remarkable performance on several datasets, which verifies the effectiveness.

**Audience:**

Yes

**Broader Impact Concerns:**

There are no further broader impact concerns.

**Claims And Evidence:**

No

**Requested Changes:**

There are several recommendations of changes here:

1. Clarify the significant discrepancy in the reported PSNR results for NAFNet on the GoPro dataset (33.02dB/33.04dB in Table 4) compared to the officially published result (33.69dB), explaining the reason for this difference. (Critical and very very improtant)
2. Redesign the ablation studies by progressively adding blocks, starting from the pseudo-sharp image, to more clearly demonstrate the incremental improvement provided by subsequent components. And conducting above ablation studies on the GoPro dataset to further enhance the overall convincingness of the analysis (Recommended)
3. Demonstrate the performance of the lightweight model on datasets expect RealBlur, particularly in comparison to other lightweight methods. (Recommended)
4. Clearly discuss and clarify the differences between the proposed framework and existing multi-stage prior arts, such as MPRNet, to highlight the novelty and differences of the proposed method. (Recommended)

**Strengths And Weaknesses:**

There are several strengths:

1. The paper is well writen. The motivation and methodology of the paper is clear.


There are several concerns:

1. The convincement of the experimental results is expected to be further verfied. Specifically, a difference exists in the reported performance of NAFNet on the GoPro dataset. The authors report PSNR values of 33.02dB and 33.04dB in Table 4, which are notably lower than the officially reported result of 33.69dB (PSNR of proposed method is 33.60). Considering NAFNet is one of the most important baselines in the paper, authors are highly expected to clarify the reason for this difference.
2. The ablation studies are not convincing. It is recommended that the ablation studies be designed to progressively add blocks, starting from the results obtained solely from the Pseudo-Sharp Image processing. It would more clearly demonstrate the incremental improvement provided by the subsequent components building upon the Pseudo-Sharp Image. Furthermore, conducting ablation studies on the GoPro dataset is encouraged to enhance the overall convincingness of the analysis
3. It may not be appropriate to compare the scalability of the proposed method at large scales, as the method does not introduces extra new information and this step-by-step strategy are usually learned implicitly by large-scale models. However, for lightweight models, this step-by-step or multi-stage methods demonstrate better performance, likely due to the introduction of prior knowledge, and it is therefore requested that the authors demonstrate the performance of the lightweight model on datasets other than RealBlur, particularly in comparison to other lightweight methods.
4. Several baselines are further expected to be clarified. I think this paradigm of the paper is pretty similar to multi-stage. The authors are encouraged to further discuss the differences from prior arts, such as MPRNet [CVPR 2021] to clearly highlight the novelty and differences of their proposed framework.

---

> ### Author Response · Authors · 2025-06-07
> **Official Comment by Authors (1/2)**
>
> Dear Reviewer 2gAJ,
>
> We sincerely thank the reviewer for your valuable comments and efforts. The insights have helped us clarify our contributions. We have revised our draft (highlighted in blue) in response to the insightful and valuable comments. We do our best to clarify each of the concerns carefully as follows:
>
> ----
> $$\textbf{(W1) Explain the gap between reported and official NAFNet PSNR on GoPro.}$$
>
> Thank you for pointing this out. In Table 4, we initially reported the results of NAFNet and RefDeblur without applying the Test-time Local Converter (TLC) [1], in order to ensure a fair baseline comparison, as TLC introduces additional computational overhead and is not universally applicable across all methods. To address the reviewer's concern, we have evaluated our models with TLC. The results are as follows:
>
> - The PSNR of NAFNet-64 improves from 33.04 dB to 33.69 dB, aligning with the officially reported performance.
> - Similarly, NAFNet-96 improves from 33.02 dB to 33.66 dB.
> - Importantly, our RefDeblur-B also benefits significantly from TLC, with its performance increasing from 33.60 dB to 34.10 dB.
>
> We have updated the results in the revised draft. Please refer to Table 4 and its discussion (Section 4.4) in the revised draft.
>
> ----
> $$\textbf{(W2) Redesign the ablation study to progressively adding components from the pseudo-sharp image in the GoPro dataset.}$$
>
> Thank you for the constructive suggestion. Following the reviewer’s recommendation, we have conducted the ablation study on the GoPro dataset to enhance the reliability of our ablation study.
> Specifically, we begin our ablation study with the pseudo-sharp image generated from NAFNet-16, which yields a PSNR of 31.82 dB. When we introduce reference training, i.e., our kernel-based method, to explicitly generate a reference image, we observe a drop in PSNR to 30.16 dB due to the presence of artifacts in the intermediate reference image. However, the reference image still contains rich texture information as exemplified in Fig.1 (c), which is not captured in the pseudo-sharp image, as exemplified in Fig.1 (b). In the final stage, applying our artifact-free training, i.e., our kernel-free method with the reference image, allows the model to eliminate artifacts while retaining fine details. This leads to a significant PSNR improvement to 33.20 dB, notably exceeding the performance of the official NAFNet-32 model (32.85 dB) under the same computational cost (around 16 GMACs). This result clearly demonstrates the effectiveness of our two-stage design, where leveraging a high-detail but flawed reference image leads to better final deblurring results when performing artifact-free training. We have updated our draft to discuss this. Please refer to Section 5 (e.g., Analyzing the role of each training stage) and Table 5 of the revised draft.
>
> |                                | Pseudo-Sharp Image | Reference Image | Final Deblurred Image |
> |--------------------------------|---------|---------|---------|
> | Pseudo-Sharp Image             | ✓       | ✓       | ✓       |
> | Reference Training             |         | ✓       | ✓       |
> | Artifact-Free Training         |         |         | ✓       |
> | **PSNR (dB)**                  | **31.82** | **30.16** | **33.20** |
>
> ----
> $$\textbf{(W3) Demonstrate the performance of the proposed method under the lightweight conditions on datasets other than RealBlur.}$$
>
> We appreciate the reviewer’s insightful comment. We agree that the most compelling test setting for our step-by-step strategy is the lightweight regime, where explicit prior guidance can compensate for limited model capacity. To address the reviewer’s concern, we evaluate RefDeblur-T (16 GMACs) on the GoPro dataset and compare it against other lightweight baselines such as NAFNet-32 and FFTFormer-16.
> As shown in the table below, the results show gains of approximately 0.4 dB PSNR over other efficient baselines such as NAFNet-32 and FFTformer-16 on GoPro dataset, confirming that the reference-guided deblurring is especially beneficial when model capacity is constrained (e.g., small). We believe that these results clearly demonstrate the advantage of our method in efficient deblurring scenarios. We have updated our draft to discuss this. Please refer to Section 5 (e.g., Efficient deblurring training) and Table 6 of the revised draft.
>
> | Method                          | GMACs | PSNR ↑ | SSIM ↑ |
> |---------------------------------|--------|--------|--------|
> | NAFNet-32    		   | 16.25  | 32.85  | 0.960  |
> | FFTFormer-16		   | 16.41  | 32.81  | 0.959  |
> | RefDeblur-T (**ours**)          | 16.33  | 33.20  | 0.967  |

---

> ### Author Response · Authors · 2025-06-07
> **Official Comment by Authors (2/2)**
>
> ----
> $$\textbf{(W4) Clarify the differences from MPRNet and highlight the novelty of the proposed framework.}$$
>
> Thank you for the valuable comment. Building on the concept of multi-stage training, our method pursues a fundamentally different objective compared to existing methods such as MPRNet. MPRNet aims to recover sharp images from blur images through three sequential refinements, but it does not explicitly target efficiency. In contrast, our pipeline assigns distinct roles to each stage. We generate an image-level reference in the first stage, and leverages it as a strong input in the second stage to build an efficient deblurring model. This reference-guided design enables us to reduce the model size and computational cost without sacrificing performance, while MPRNet is designed to use several heavy multi-stage refinements for better reconstruction. In summary, unlike MPRNet "performs heavy multi-stage refinements", our method "decouples generation and correction". We first create a strong but imperfect reference, and then efficiently correct its artifacts. We have included this discussion in Section 3.4 (e.g., Comparison with MPRNet) of the revised draft.
>
> ----
> $$\textbf{References}$$
>
> [1] Chu et al., Revisiting global statistics aggregation for improving image restoration, ECCV 2022

---

> > ### Comment · Reviewer_2gAJ · 2025-07-04
> >
> > Dear authors,
> >
> > Many thanks for your kind response. I carefully read the reponse, I think my concerns are well addressed, especially the first concern.
> >
> > best,
> > Reviewer 2gAJ

---

> > > ### Author Response · Authors · 2025-07-04
> > > **Official Comment by Authors**
> > >
> > > Dear Reviewer 2gAJ,
> > >
> > > We appreciate the reviewer’s valuable feedback and insightful suggestions. Also, we are glad to hear that the reviewer's concerns are well addressed. The valuable and constructive feedback has helped us clarify our contribution. Thank you once again for the thoughtful review and the opportunity to refine our work. Please feel free to reach out with any additional questions or suggestions.
> > >
> > > Best regards,
> > >
> > > Authors

---

### Review · Reviewer_6jES · 2025-05-25

**Summary Of Contributions:**

This paper works on the problem of blind motion deblur, where the motion blur kernel is unknown. Instead of either estimate the spatial-variant non-uniform kernel precisely or skip the kernel estimation and only predict the deblurred image in a single forward pass, the contribution of this paper is a two-stage strategy. First, a reference image deblurred from off-the-shelf networks and the blurred input images are transformed into frequency domain and a uniform kernel is estimated from the frequency image via a kernel estimation network. Then the authors deblur the blurred input image using this estimated kernel and then predict a residual image which is added to the kernel deblurred image as the final clean image. To validate the contribution of this two-stage strategy, the authors conduct experiments on various dataset and demonstrate improvement on both performance and efficiency.

**Audience:**

Yes

**Broader Impact Concerns:**

This paper is about motion deblur which is a low-level image processing problem, therefore these is no concern about ethical issues.

**Claims And Evidence:**

Yes

**Requested Changes:**

- First, Please refer to the weakness.
- Second, I would suggest the author to run the proposed pipeline iteratively, where the reference image is the output from the last iteration and see how the performance improves or diverges.

**Strengths And Weaknesses:**

Strengths:
- The paper itself is generally well organized, with comprehensive introduction to the problem and motivation.
- The method design is simple and effective.
- The notation in this paper is clear and easy to follow.
- The method achieves a good balance between performance and efficiency.

Weakness:
- The paper should elaborate more about why estimate spatial-variant kernel is less effective than the two-stage pipeline proposed in the paper. For example, the authors should vary the network capacity of spatial-variant kernel estimation and see how the performance increase along with network capacity.
- The kernel estimated in this paper is not guaranteed to be physically valid. There is no constraint enforce the element of the kernel to be positive and sum equals to one. The authors should discuss and proof that this constraint is unnecessary.
- The paper requires an external deblur network to generate the reference image, it is unclear whether the computational of the external deblur network is included in the computation cost. In addition, the authors did not compare with different external network to generate the reference. Since the kernel estimation network is trained on ground-truth image and there is a potential training-inference gap, the authors should validate different external models rather than leaving it to future work.
- All the evaluation metrics in the paper are PSNR and SSIM, more advanced perceptual metric, at least LPIPS is necessary.

---

> ### Author Response · Authors · 2025-06-07
> **Official Comment by Authors (1/2)**
>
> Dear Reviewer 6jES,
>
> We sincerely thank the reviewer for your valuable comments and efforts. The insights have helped us clarify our contributions. We have revised our draft (highlighted in blue) in response to the insightful and valuable comments. We do our best to clarify each of the concerns carefully as follows:
>
> ----
> $$\textbf{(W1) Comparison with spatially-variant kernel estimation under different network capacities.}$$
>
> We appreciate the reviewer's suggestion to further elaborate on the comparison between spatially-variant kernel estimation and our proposed two-stage pipeline. Deblurring with blur kernels, by closely aligning with physical blur modeling, facilitates producing texture-rich sharp images. However, estimating spatially-variant blur kernels inherently struggles with the highly ill-posed inverse problem, leading to significant training difficulties. This is because many different per-pixel kernel combinations could potentially explain the blur input, leading to ambiguities and inaccurate kernel results. While increasing network capacity for a spatially-variant kernel estimation model might yield some performance gains, these gains are limited by these fundamental challenges (see the results in the table below). On the other hand, our two-stage pipeline is designed to leverage the strengths of both kernel-based and kernel-free deblurring methods. Crucially, it adopts a uniform kernel estimation in its initial stage, thereby inheriting the capacity to render rich-detailed reference image (but with some errors), and then incorporating a kernel-free method, specialized in correcting intricate and spatially-variant remaining errors, to effectively removing complex blur degradations while preserving the initial texture details within the reference image. As demonstrated in the table, our proposed two-stage pipeline demonstrates scalability with increased network capacity and achieves superior performance compared to spatially-variant kernel estimation. We have updated our draft to discuss this. Please refer to Section 5 (e.g., Spatially-variant kernel estimation vs. our hybrid deblurring framework) and Table 7 in the revised draft.
>
> | Method                             | RefDeblur-T | RefDeblur-S | RefDeblur-B |
> |------------------------------------|-------------|-------------|-------------|
> | Spatially-Variant Kernel Estimation | 31.24       | 31.42       | 31.56       |
> | Hybrid Deblurring Framework        | 32.70       | 33.38       | 33.81       |
>
> ----
> $$\textbf{(W2) Discuss why the estimated kernel does not impose physical constraints and explain why such constraints are unnecessary.}$$
>
> We acknowledge the reviewer's comment regarding the physical validity of the estimated blur kernel. We opted not to enforce strict physical constraints (non-negativity and sum-to-one) on the blur kernel because such rigid constraints could introduce training difficulty or potentially hinder the network's ability to find an optimal reference image. Instead, we applied a [0, 1] clamping operation to the generated reference image. This clamping scheme serves two key purposes: 1) It ensures that the generated reference image adheres to the valid pixel value constraints, making it a stable input for the subsequent kernel-free model. 2) More importantly, it acts as an implicit regularization during the reference training. For example, by clamping the reference pixel values to the [0,1] range, it implicitly steers our kernel-based model towards generating physically-valid blur kernels without strong physical constraints (e.g., non-negativity and sum-to-one). This suggests that the use of such implicit regularization in generating the reference image outweighs the potential drawback of the blur kernel not being strictly physically valid, as it facilitates easier and more stable reference training. We have updated our draft to discuss this. Please refer to Section 3.4 (e.g., Blur kernel constraints) of the revised draft.

---

> ### Author Response · Authors · 2025-06-07
> **Official Comment by Authors (2/2)**
>
> ----
> $$\textbf{(W3) Clarify that the external deblurring network's cost is included in the total computational cost, and validate the impact of using different external networks for reference generation.}$$
>
> Thank you for the constructive comment. Regarding computational cost, the external deblur network's GMACs are included in our total computational cost as reported in Table 8 of the revised draft; for instance, RefDeblur-S's 62.61 GMACs contains 16.25 GMACs for generating a pseudo-sharp image.
> We opted to train our kernel-based model with Ground-Truth (GT) sharp images for two primary reasons. Firstly, its final deblurring performance was comparable to training directly with pseudo-sharp images. More importantly, this GT-based training strategy provides crucial extensibility: it allows us to utilize pseudo-sharp images generated by any off-the-shelf deblurring model without requiring retraining of our kernel-based model.
> To validate this extensibility and demonstrate the impact of pseudo-sharp image quality (also suggested by the reviewer), we conduct additional experiments using diverse pseudo-sharp images generated by external deblurring models, including NAFNet, FFTFormer, and Restormer. As shown in the table below, these results show that employing higher-quality pseudo-sharp images yields better final deblurring performance. This directly demonstrates that leveraging stronger pseudo-sharp images helps mitigate the potential training-inference gap (e.g., training with GT sharp images, but evaluating with pseudo-sharp images), leading to better final deblurring performance. We have updated our draft to discuss this. Please refer to Section B.2 and Table 10 of the Appendix in the revised draft.
>
> | Method                             | RefDeblur-B |
> |------------------------------------|-------------|
> | Blur + Pseudo-Sharp (NAFNet)      | 33.81       |
> | Blur + Pseudo-Sharp (FFTFormer)      | 33.22       |
> | Blur + Pseudo-Sharp (Restormer)      | 32.88       |
>
>
> ----
> $$\textbf{(W4) Add LPIPS metric.}$$
>
> Thank you for the valuable suggestions. We have added the LPIPS metric to Tables 2 (RealBlur) and 4 (GoPro) to compare the perceptual quality across methods in the revised draft. Note that we evaluate LPIPS using publicly available pre-trained models. However, for the RSBlur dataset, the pre-trained models are not released, making it infeasible to report LPIPS results for this dataset. As shown in Table 2 and 4, the results show that our proposed two-stage pipeline achieves superior perceptual quality compared to other methods. This suggests that the reference image utilized in our method plays a crucial role in enhancing perceptual quality, thereby leading to better LPIPS scores. We have included this discussion in Section 4.3 (e.g., Quantitative and qualitative results) and 4.4 (e.g., Results on GoPro) of the revised draft.
>
> ----
> $$\textbf{(W5) Run the proposed pipeline iteratively.}$$
>
> Thank you for your insightful suggestion to explore an iterative evaluation of our pipeline, where the output from one iteration serves as the pseudo-sharp image for the next. Following the reviewer's recommendation, we iteratively evaluate our pipeline. However, as shown in the table below, we observe that performance tends to degrade with subsequent iterations. We believe this is because the pseudo-sharp images generated from previous iterations differ in characteristics from those the kernel-free model was originally trained with. This can lead to 'unseen' input for the next iteration, leading to performance degradation. Therefore, while the concept of iterative refinement is appealing, in our current framework, using the iteratively generated pseudo-sharp images appears to negatively impact the final performance.
> To further investigate this, we perform an additional experiment where the kernel-free model is retrained for the second iteration. In this setup, we use the deblurred images from the first iteration as new pseudo-sharp images for the second. Interestingly, this retrained kernel-free model shows improved performance. This result supports our assumption that the initial performance drop is caused by the second kernel-free model encountering out-of-distribution pseudo-sharp images when reused iteratively without retraining. We have updated our draft to discuss this. Please refer to Section B.5 and Table 13 of the revised draft.
>
> | Iterative evaluation 	  | PSNR      | SSIM        |
> |--------------------------------|------------|-------------|
> | 1st iteration		  | 32.70       | 0.927       |
> | 2nd iteration     		  | 32.62       | 0.924       |
> | 3rd iteration		  | 32.57       | 0.924       |
> | 2nd iteration (retraining)      | 32.83       | 0.928       |

---

### Decision · Action_Editor_qyHu · 2025-08-01

**Recommendation:** Accept as is

**Audience:**

Yes

**Audience Explanation:**

The paper is of interest for researchers working on ML for image restoration.

**Claims And Evidence:**

Yes

**Claims Explanation:**

The propose a novel model for blind motion deblurring. They claim performance comparable to the state-of-the-art with reduced computational costs. The authors provide evidence for these clames by conducting experiments on various datasets.